# Suppression of bacterial cell death underlies the antagonistic interaction between ciprofloxacin and tetracycline

James Broughton [ID][1], Achille Fraisse [ID][1] & Meriem El Karoui [ID][1,2]✉

## Abstract

Antibiotic combinations aim to maximise drug treatment efficiency and minimise resistance evolution, but a full understanding of their effect on bacterial cells is lacking. The interaction between the DNA-damaging antibiotic ciprofloxacin and the translation inhibitor tetracycline is antagonistic, resulting in a weaker effect on bacterial growth than expected from each drug individually. While this antagonism has been analysed at the population level, it has not been investigated at the single-cell level. We used a microfluidic device to quantify the antagonism between ciprofloxacin and tetracycline in single bacterial cells under three nutrient conditions. Improved growth results from increased survival of cells under the drug combination compared to ciprofloxacin alone. This effect depends on the initial drug-free growth rate, with better suppression in nutrient-rich conditions. Quantifying the DNA damage response (SOS response) revealed two sub-populations among cells that died upon ciprofloxacin treatment. The larger low-SOS sub-population, which showed increased survival compared to high-SOS cells, explains the stronger antagonistic effect in nutrient-rich conditions. Our results underscore the importance of single-cell quantification in understanding bacterial responses to antibiotic combinations.

**Keywords** Antibiotic Combinations; Antagonism; Growth-dependence; Single-cell; SOS Response
**Subject Category** Microbiology, Virology & Host Pathogen Interaction

## Introduction

Our society is facing a major crisis with the rise of antimicrobial resistance (Murray et al, 2022). As the pace of the discovery of new antimicrobial molecules has slowed down, it is crucial to use current molecules to their full potential. One strategy to maximise the efficiency of antibiotic treatment and minimise the evolution of resistance is to use antibiotic combinations (Chait et al, 2007; Tyers and Wright, 2019). Such combinations can impact bacterial growth differently depending on their interaction: as the antibiotics interact, their effect on bacterial growth can be stronger (synergistic) or weaker (antagonistic) than expected from the additive effect of each single drug if they act independently (Mitosch and Bollenbach, 2014; Roemhild et al, 2022; Yeh et al, 2006). Antibiotic interactions are determined quantitatively by comparison to this expected additive effect which can be calculated using two models, Bliss independence (Bliss, 1939) and Loewe additivity (Loewe, 1928, 1953), depending on how the data is collected. Although antibiotic interaction networks have been systematically determined (Chevereau and Bollenbach, 2015; Yeh et al, 2006), the mechanism underlying many interactions have yet to be found or fully quantified.

The interaction between DNA-damaging antibiotics (such as fluoroquinolones) and some translation inhibitors, such as chloramphenicol or tetracycline, has been analysed in detail at the population level. This interaction is antagonistic (i.e., effect weaker than expected) and, in some concentration regimes, can even be suppressive, i.e., the drug interaction results in a higher population growth rate than that observed for cells exposed to one of the drugs alone (Bollenbach et al, 2009). The interaction is also directional: the effects of the translation inhibitor impedes the activity of the DNA-damaging drug, but not vice versa (Bollenbach et al, 2009; Roemhild et al, 2022). The current understanding of this suppressive interaction is based on combining the known bacterial response to each antibiotic. Fluoroquinolones (such as ciprofloxacin) are bactericidal antibiotics that cause cell death through the inhibition of the activity of the type II topoisomerases. This leads to the accumulation of DNA double-strand breaks (DSBs) and interferes with chromosome replication and transcription (Bush et al, 2020). In *Escherichia coli* the DNA damage response, called the SOS response, is regulated by the transcriptional repressor LexA. DSBs are processed to form a single-stranded DNA region onto which the RecA protein is loaded. This leads to the formation of a RecA nucleoprotein filament, which triggers self-cleavage of LexA and the expression of approximately 50 genes important for DNA repair, DNA replication, mutagenesis and cell survival (Kreuzer, 2013; Mo et al, 2016). Strong induction of the SOS response leads to cell filamentation through expression of the division inhibitor *sulA* (Huisman and D'Ari, 1981). SOS response

[1]Institute of Cell Biology, School of Biological Sciences, University of Edinburgh, Edinburgh, UK. [2]LBPA, Ecole Normale Supérieure - Paris-Saclay, CNRS UMR 8113, Gif-sur-Yvette, France. ✉E-mail: meriem.elkaroui@ed.ac.uk

expression and filamentation is commonly associated with sub-MIC ciprofloxacin treatment (Bos et al, 2015; Butler et al, 2023; Pribis et al, 2019). Induction of the SOS response is also linked to the activation of several toxin-antitoxins (Yamaguchi and Inouye, 2011), some of which have been shown to disrupt bacterial growth and lead to cell death (Prysak et al, 2009; Su et al, 2022; Unoson and Wagner, 2008). In contrast, translation inhibitors such as tetracycline are bacteriostatic and lead to reduced growth by inhibiting ribosome function and, therefore, protein production (Barrenechea et al, 2021; Chopra and Roberts, 2001; Poehlsgaard and Douthwaite, 2005). It has been proposed that under fluoroquinolone treatment, the cells do not appropriately downregulate ribosomal expression resulting in a skewed protein-DNA ratio due to excess and metabolically costly protein production (Bollenbach et al, 2009). When combined, the translation inhibitor partially restores the imbalance in the protein-DNA ratio leading to improved population growth (Bollenbach et al, 2009).

Understanding how drug combinations are affected by changing nutrient environments is important for treating bacterial infections. Indeed, bacterial growth rates are known to vary depending on where in the body the infection is located. For example, *E. coli* doubling times can range from 3 h in the intestine to ~22 min in the urinary tract (Forsyth et al, 2018; Myhrvold et al, 2015). The composition of bacterial cells is known to respond to changes in the nutrient composition of the medium according to universal constraints on their proteome (Hui et al, 2015; Scott et al, 2010). This can affect the bacterial response to ribosome-targeting antibiotics: bacterial susceptibility to tetracycline is dependent on their drug-free growth rate (Greulich et al, 2015) with fast-growing bacteria more affected than slow-growing ones. In addition to changes in cell proteome composition, changing growth rates by varying nutrient composition can impact DNA replication and potentially the efficiency of DNA repair. Fast-growing cells (less than 60 min doubling time) undergo multi-fork DNA replication by initiating multiple overlapping rounds of replication (Cooper and Helmstetter, 1968). This results in higher DNA content per cell, a higher number of replication forks and multiple partial chromosomal copies in fast-growing cells. Because of their higher DNA content, treatment with ciprofloxacin may lead to more DSBs in fast-growing cells, but these cells also have a higher number of potential copies for homologous DNA repair, while slower-growing cells may be limited in their capacity to repair due to a lack of homologous copies of the chromosome. This may lead to growth dependence of the susceptibility to ciprofloxacin, which has indeed been shown at the population level (Smirnova and Oktyabrsky, 2018). Growth dependence (here growth dependence refers to change in growth rate due to change in nutrient composition of the medium without antibiotic) of susceptibility to individual antibiotics suggests that antibiotic combinations are also likely affected by changes in nutrient quality, but this has not yet been investigated, particularly at the single-cell level.

It is well-established that isogenic bacterial cultures show high phenotypic heterogeneity at the single-cell level, including cell-to-cell variability in gene expression, growth rates, stress responses, and susceptibility to antibiotics (Brandis et al, 2023; Choudhary et al, 2023; Jones and Uphoff, 2021; Sampaio et al, 2022; Vincent and Uphoff, 2021). Antibiotics can induce further heterogeneity in the population. For example, we showed previously that DNA damage leads to high heterogeneity in division rates due to

variability in SOS expression among single cells thus impacting the population dynamics (Jaramillo-Riveri et al, 2022). Single-cell elongation rates and division rates can also become uncoupled under DNA stress in cells that filament, leading to inaccurate optical density readings (Stevenson et al, 2016) and growth rate measurements (Jaramillo-Riveri et al, 2022). Therefore, acquiring single-cell time-resolved data is vital for understanding the dynamic response of bacteria to antibiotic combinations.

Traditionally, population growth rates are determined using bulk-level methods (e.g., optical density or colony-forming unit counting), which measure net growth and population-averaged behaviour. These measurements are a function of cell mass accumulation (which correlates with elongation for rod-shaped bacteria), cell birth/division, and cell death. For the antagonistic drug combination between ciprofloxacin and tetracycline, it is unclear what effect the drug combination has on each of these parameters and which one results in the observed increase in population growth rate. For instance, faster growth than expected under the drug combination could indicate an increase in single-cell elongation, division, survival, or a combination thereof. In this study, we used a microfluidic device called the mother machine (Jaramillo-Riveri et al, 2022; Wang et al, 2010) to measure each of these parameters separately and determine how the drug combination affects bacteria in three growth conditions. We showed that the antagonism between ciprofloxacin and tetracycline is due to increased survival of cells under the drug combination compared to ciprofloxacin alone and is not due to increased rates of division or mass accumulation. This effect is growth rate-dependent, with better suppression in fast growth than in slow growth conditions when this is modulated by nutrient quality. Quantification of the DNA damage response revealed two sub-populations among the cells dying upon ciprofloxacin treatment, with some cells reaching a very high level of SOS while others had a lower level of SOS similar to cells that survived. The low-SOS cells were more frequent in fast growth conditions and showed increased survival in the antibiotic combination contrary to the high-SOS ones, thus explaining the growth dependence of the antagonistic effect of tetracycline on ciprofloxacin.

## Results

### The single-cell response to sub-lethal concentrations of ciprofloxacin and tetracycline is highly heterogeneous

To determine the conditions where the combination of ciprofloxacin (CIP) and tetracycline (TET) is antagonistic (or suppressive), we performed an initial characterisation in bulk culture using a defined medium that supports fast growth (M9-based medium supplemented with glucose and amino acids, here referred to as glu-aa). Analysis of growth rate using $OD_{600}$ measurements confirmed the suppressive interaction of TET and CIP in a concentration range of 4.5–13.5 ng/mL for CIP and 0.2–0.6 µg/mL for TET (Appendix Fig. S1A,B, see Appendix Methods and Appendix Fig. S2 for experimental details). Initial tests in a microfluidic device called the 'mother machine', which allows the monitoring of hundreds of single-cell lineages under defined growth conditions (Wang et al, 2010), revealed that when exposed to 5 ng/mL CIP, cells undergo high levels of filamentation.

Filamentation under CIP treatment is expected since CIP creates DNA damage and leads to the induction of cell division inhibitors as part of the SOS response (Kreuzer, 2013). However, the observed level of filamentation resulted in an unacceptably-large loss of cell lineages from the microfluidic device. We, therefore, used 3 ng/mL CIP and 0.2 μg/mL TET for all the subsequent experiments: this concentration allows the detection of the antagonistic relationship between the two antibiotics (evaluated in bulk in three different growth media, Appendix Fig. S3 and Appendix Table S1) while limiting excessive cell filamentation, which facilitates imaging of individual cells in the microfluidic device.

We first sought to evaluate the single-cell physiological response of *E. coli* to exposure to 3 ng/mL CIP and 0.2 μg/mL TET separately. We cultivated cells in the mother machine and after adaptation measured the bacterial growth for 2 h. We then added the relevant antibiotic for 12 h, removed it, and monitored the cells for a further 10 h to allow recovery from antibiotic treatment (Fig. 1A). The bacteria were imaged by fluorescence microscopy and segmented and tracked automatically (see 'Methods'). We only tracked the lineage of the cell at the end of the microfluidic microchannel (the 'mother' cell). We measured the single-cell elongation rates, cell length, and gene expression using a constitutively expressed gene integrated on the chromosome ($P_{tetO1}$-*mKate*2). Additionally, the strain carried a transcriptional reporter for the SOS response ($P_{sulA}$-*mGFP*) (Jaramillo-Riveri et al, 2022). As shown in example kymographs (Fig. 1B), cells in the no-drug condition grew and divided for the entire duration of the experiment. When exposed to TET, the cells became slightly smaller, but continued to elongate and divide (Fig. 1C), as expected when cells are exposed to sub-lethal concentrations of a bacteriostatic antibiotic. In contrast, when the bacteria were exposed to CIP, we detected a high level of heterogeneity in their response. Some cells continued growth and division, whereas others filamented to varying degrees before either resuming division (Fig. 1D) or stopping growth and division for the remainder of the experiment (Fig. 1E). Additionally, some cells ceased growth and division without filamenting.

We then quantified single-cell elongation rates for all cells under CIP, TET, and the no-drug control. Elongation rate trajectories under drug-free conditions remained stable throughout the experiment (median line, Fig. 1F). Under TET treatment, elongation rates initially decreased before levelling off to a median level ~16% below pre-antibiotic treatment. When TET was removed, elongation rates recovered to pre-antibiotic levels (Fig. 1G). To compare the elongation rates over time, we plotted distributions drawn from sequential two-hour periods under different antibiotic treatments for the duration of the experiment. The response appeared homogeneous across the population in the no-drug and TET conditions, suggesting that there was limited population heterogeneity under these conditions (Fig. 1I,J). Under these conditions, we detected a very small population that ceased growth toward the end of the experiment, suggesting age-related death as has been reported before (Wang et al, 2010; Yang et al, 2019).

In contrast, elongation rates under CIP treatment were highly heterogeneous (Fig. 1H). A large fraction of the cells did not show any change in elongation rates during CIP exposure. However, a second population of cells stopped elongation entirely (trajectories near 0 h$^{-1}$; Fig. 1H). We did not observe any significant recovery of cells' elongation within the 10 h of drug-free growth post antibiotic

treatment, suggesting that these cells had died. This second population appeared clearly in the distributions shown in Fig. 1K from the 4–6 hour window (i.e., 2 to 4 h after antibiotic exposure), and its proportion increased over time. To further check whether these cells were indeed dying, we measured the expression of the constitutively expressed mKate2 marker: all showed an exponential decrease in fluorescence after elongation stopped, consistent with arrest of protein production and bleaching of the fluorophore, indicating that these cells were not metabolically active (Appendix Figs. S4–S6F). We also checked whether a longer recovery period of 36 h would lead to some cells resuming growth, and found no significant recovery. Taken together, these results indicate that cells that have abruptly stopped elongation are 'dead', similar to the definition used in previous studies (Robert et al, 2018; Vincent and Uphoff, 2021). Therefore, we conclude that even at the very low concentration of 3 ng/mL CIP (whereas the MIC is typically 12 ng/mL (Bollenbach et al, 2009; Pribis et al, 2019)) we observe a significant amount of cell death. This is not surprising given that CIP is bactericidal and has been previously observed at the population level (Coates et al, 2018).

## The CIP-TET combination is suppressive due to improved survival of bacterial cells

Single-cell elongation rates under the drug combination uniformly decreased similarly to cells treated with TET alone (Fig. 2A,B). Median elongation rates of the growing population at the end of drug treatment decreased by ~14% (versus 16% in TET alone). However, in contrast to cells exposed to CIP, we observed a marked decrease in the proportion of cells that stopped growth and division under the drug combination (Fig. 2C). Moreover, we did not observe an increase in elongation rates (when compared to TET exposure alone) for the growing population when TET was added to CIP (Fig. 2C, Table EV2). This indicates that, at the single-cell level, there is no suppressive effect on elongation rates under the drug combination, but that the drug combination acts by decreasing the number of cells that die. Therefore, the increase in population growth rate that we observed in bulk experiments (Appendix Figs. S1 and S2) and has been reported before (Bollenbach et al, 2009), is not due to better growth (i.e., increase in mass/volume of cells) in CIP-TET but due to lower cell death under the antibiotic combination.

## The CIP-TET combination is suppressive in all growth conditions

To check whether survival was affected by drug-free growth rate, we compared the survival of bacteria under CIP, TET, and CIP-TET exposure in three different growth conditions using the same concentration as before. In addition to the glu-aa condition, we used an M9-based medium supplemented with glucose (referred to as 'glu') or glycerol (referred to as 'gly'). As expected, the median elongation rate was higher in glu-aa than in glu and gly (1.13, 0.54, and 0.28 h$^{-1}$ median elongation rates, respectively) in the no-drug control (See Appendix Figs. S7 and S8 for data representing the glu and gly growth medium, respectively).

To quantify the survival of individual cells more precisely, we devised a simple algorithm to classify a lineage as having survived or died (Appendix Fig. S9, see Appendix Methods for details). Following

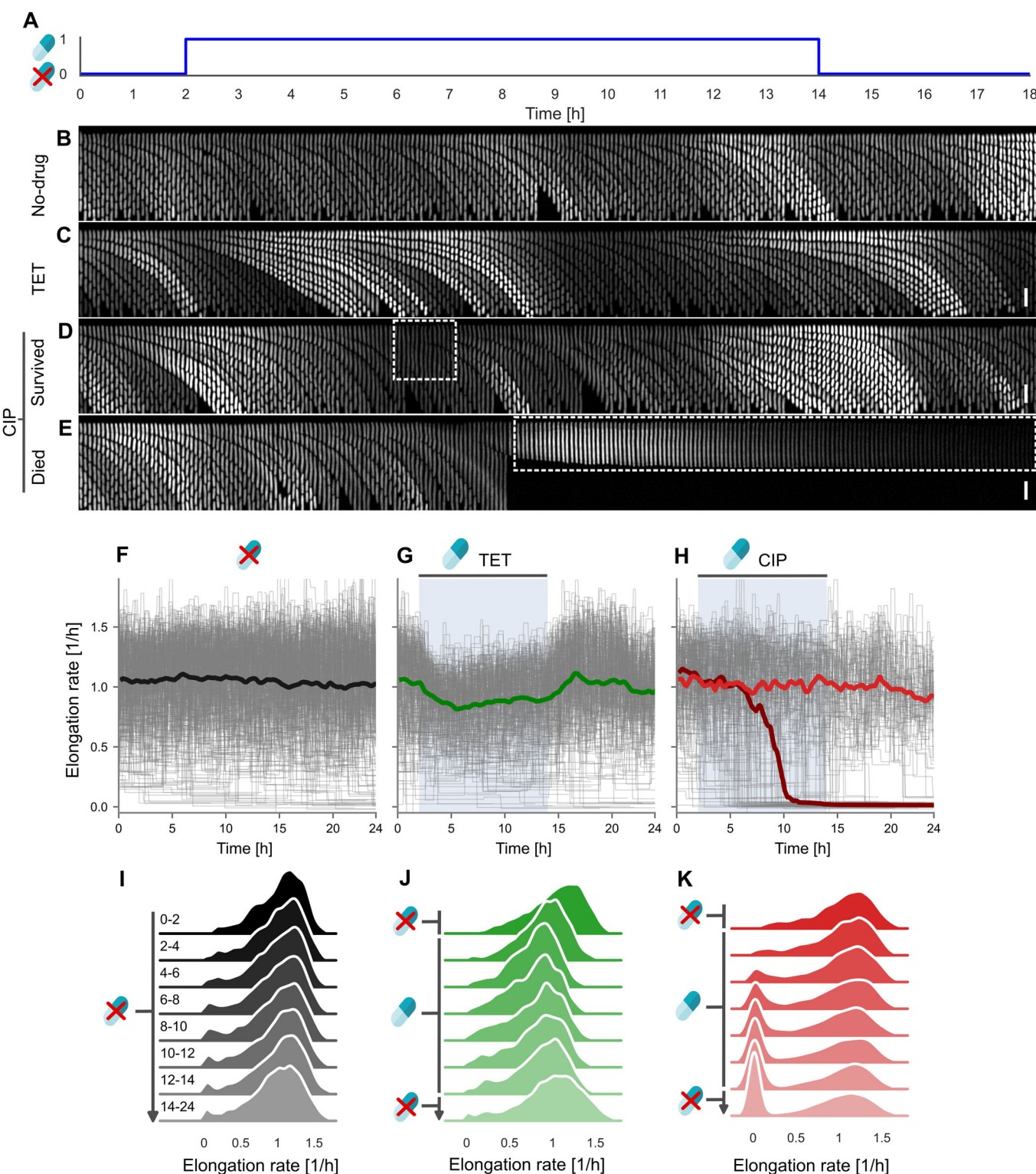

the classification of lineage fate, we generated survival curves (Fig. 3A–C) using the Kaplan–Meier (K-M) estimator (Kaplan and Meier, 1958). As expected, most cells survived in the no-drug control and in the TET treatment. Similar to our initial observation in glu-aa, we observed a small amount of death in glu and in gly at the end of the

experiment, likely due to ageing (Wang et al, 2010; Yang et al, 2019). In contrast, under CIP treatment, we observed reduced survival in all growth conditions: cells died during antibiotic treatment, or within a few hours after antibiotic removal, probably due to residual antibiotic or residual DNA damage in the cells. After the antibiotic was removed,

◀ **Figure 1. Single-cell response under sub-lethal antibiotic treatment is heterogeneous.**

(A) Schematic indicating when antibiotic was introduced (hour 2) and then removed (hour 14). (B–E) Example kymographs showing growth in the glu-aa medium under no-drug, 0.2 µg/mL tetracycline and 3 ng/mL ciprofloxacin treatment. Shown is fluorescent intensity from the $P_{tetO1}$-mKate2 constitutive reporter. Under CIP treatment, some cells continued growth and division with some filamentation (D, white box), while others stopped division (E, white box). Kymographs were constructed as a montage of frames with 5-min intervals from a single microchannel. Scale bar represents 5 µm. (F–H) Single-cell trajectories of elongation rates under no-drug (F, $n = 334$ cells), 0.2 µg/mL tetracycline (G, $n = 237$), and 3 ng/mL ciprofloxacin (H, $n = 191$) treatment. Antibiotics were introduced between hours 2–14, indicated by the blue shaded area. Grey lines represent individual mother cell lineage trajectories. Solid lines represent the median elongation rate of the population. In (H), the red solid line represents cells that survived CIP treatment and the maroon solid line represents cells that stopped growth under CIP treatment. Shown is data from one experimental replicate for growth in the glu-aa medium. See Appendix Figs. S7 and S8 for data representing the glu and gly growth medium, respectively. (I–K) Distributions of single-cell elongation rates for the no-drug control (I), TET treatment (J) and CIP treatment (K) drawn from sequential two-hour periods under different antibiotic treatments in the glu-aa growth medium. Distributions are kernel density estimates of the underlying histogram pooled from at least two experimental replicates. Time periods, in hours, are indicated to the left of each distribution. Antibiotics were introduced from hours 2–14 as indicated.

the death rate decreased. Importantly, CIP was more effective in glu-aa (fast growth condition) than in gly, in keeping with what has been observed previously at the population level (Smirnova and Oktyabrsky, 2018). Under the antibiotic combination, single-cell survival significantly improved (Log-rank test, $p < 0.001$) in all growth conditions (Fig. 3A–C, compare red and blue curves). When measuring mean survival fractions at the end of the experiment (Fig. 3D), cells under CIP showed 43%, 38% and 36% survival for gly, glu & glu-aa, respectively. Under the CIP-TET combination, the proportion of cells that had survived was 61%, 66% and 76% for gly, glu & glu-aa, respectively, which is significantly higher in all growth conditions compared to CIP alone (two-sided $t$-test, $p < 0.05$). The improvement in cell survival was higher in fast-growth compared to slow-growth conditions (see below).

We then calculated the expected survival ($Survival_{CIP}Survival_{TET} = \ln(Survival_{CIP}) + \ln(Survival_{TET})$) for the antibiotic combination under the Bliss independence hypothesis (Bliss, 1939; Demidenko and Miller, 2019) ('Methods'). In all growth conditions, we observed that the survival fraction under the drug combination is significantly larger (one-sided $t$-test, ANOVA model, $p < 0.005$) than predicted by independence ($Survival_{CIP\text{-}TET} \gg Survival_{CIP}Survival_{TET}$), which indicates that the interaction is antagonistic on cell survival. Moreover, as survival was higher than in CIP treatment alone, we conclude that the drug interaction is suppressive (Roemhild et al, 2022) in all growth conditions, even at the very low CIP concentration that we used.

## The SOS response is highly heterogeneous

In order to better understand the underlying mechanism that leads to cell death, we measured the SOS response of cells exposed to the different antibiotic treatments using the $P_{sulA}$-mGFP transcriptional reporter (Fig. 4; Appendix Fig. S10). We calculated the average fluorescence per cell area (which we refer to as "SOS expression") as a proxy for GFP concentration for each cell at each time point (Jaramillo-Riveri et al, 2022). To compare SOS expression in all conditions, we analysed the SOS expression distribution over the final two-hour window of antibiotic exposure, during which the SOS expression had reached a steady state. In the no-drug control and under TET exposure, the cells did not substantially induce SOS in any of the growth conditions. However, as expected, under CIP there was an increase in SOS expression (median SOS expression for no-drug control: 118, 95, 93 a.u. in glu-aa, glu, & gly; median SOS expression for CIP: 227, 473, 462 a.u.; Fig. 4A–C compare red and black lines). Moreover, we noticed that the SOS response was

highly heterogeneous, with a sub-population of cells reaching an SOS level ~10 times higher than the rest of the population (Fig. 4B,C, red lines) which we will refer to as the high-SOS population below (we refer to the other population as the "low-SOS" population). The high-SOS population was much more abundant in slow growth conditions (glu & gly) and hardly noticeable in glu-aa. In the CIP-TET condition, the high-SOS population, surprisingly, did not seem to be affected when compared to CIP treatment alone (Fig. 4B,C, blue lines). The low-SOS population did not show much change in SOS expression in the glu-aa medium (Fig. 4A). However, in glu and gly we observed a decrease of SOS expression in the low-SOS population as can be expected from the impact of TET on protein production (compare main peak of red and blue lines in Fig. 4B,C). Ribosome expression is known to be upregulated under TET treatment, which results in a corresponding decrease in the expression of constitutive genes due to well-established universal proteome constraints (Hui et al, 2015; Scott et al, 2010). Moreover, slow-growing cells have a higher capacity to increase their ribosome levels than fast-growing cells, resulting in a larger impact on the constitutive protein fraction. Thus, this observed growth dependence of SOS expression under CIP-TET is expected: degradation of LexA upon DNA damage leads to the de-repression of SOS promoters, which behave similarly to constitutive promoters. Protein expression from these promoters will, therefore, be affected to a larger extent in slow growth than in fast growth under TET inhibition.

To test whether the level of SOS expression could be indicative of the fate of the cells, we evaluated the SOS expression among cells that had survived or died under CIP exposure. Among the survivors, very few cells reached a high-SOS level in any of the growth conditions, as shown in Fig. 4D–F. Moreover, their SOS signal returned within a 5-hour window to their pre-antibiotic level upon removal of CIP suggesting that DNA damage had been repaired. The cells that died exhibited a markedly different behaviour: their SOS level peaked, followed by an exponential decrease of fluorescence, consistent with a stop in protein production and exponential decay due to bleaching (similar to what we observed for the constitutively expressed protein mKate). In these cells, we did observe two sub-populations, with some cells reaching a high SOS level (yellow trajectories in Fig. 4G–I) while others died after expressing a low level of SOS (magenta trajectories). We therefore decided to sub-classify the cells that died into two categories depending on their SOS response level (see Appendix Methods) (Fig. 4G–I, left column). The fraction of high-SOS cells was only 12% in glu-aa but increased to 25% in glu and

 

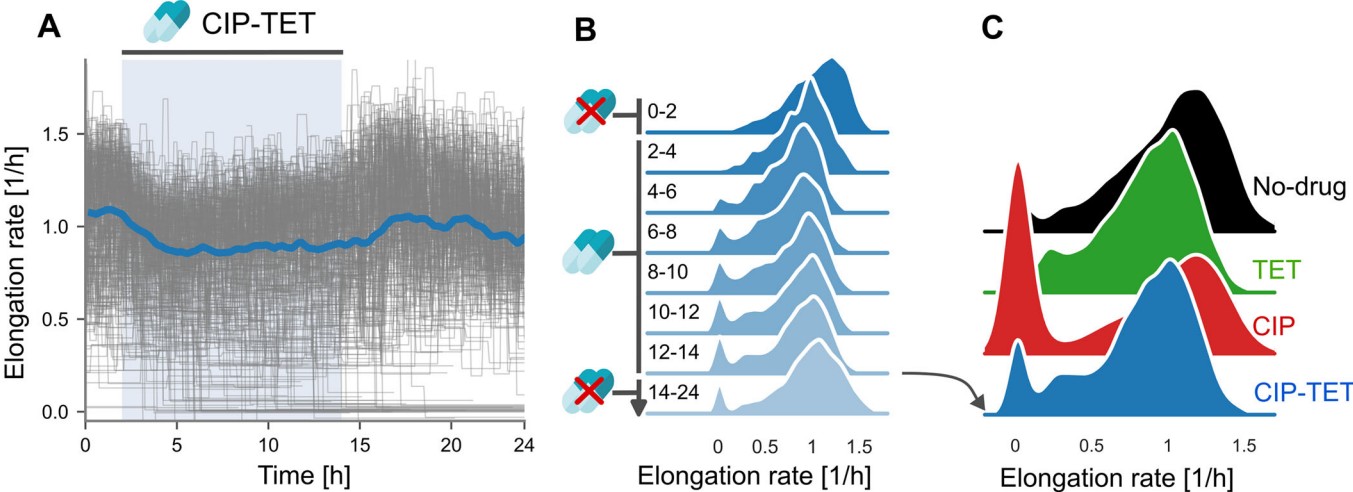

**Figure 2. The drug combination improves cell survival with no increase in single-cell elongation rates.**

(A) Single-cell trajectories of elongation rates under the combination of 0.2 μg/mL tetracycline and 3 ng/mL ciprofloxacin treatment ($n = 304$ cells). Antibiotics are introduced between hours 2–14, indicated by the blue shaded area. Grey lines represent individual mother cell lineage trajectories. Solid lines represent the median of the population. Shown is data from one experimental replicate for growth in the glu-aa medium. (B) Distributions of single-cell elongation rates drawn from sequential two-hour periods under CIP-TET treatment in the glu-aa growth medium. See Appendix Figs. S7 and S8 for data representing the glu and gly growth medium, respectively. Distributions are kernel density estimates of the underlying histogram pooled from at least two experimental replicates. Time periods, in hours, are indicated to the left of each distribution. Antibiotics were introduced from hours 2–14 as indicated. (C) Distributions of single-cell elongation rates under different treatment conditions during the final 2 h of antibiotic treatment. As in (B), distributions are kernel density estimates of the underlying histogram pooled from at least two experimental replicates.

30% in gly, suggesting that in these conditions a large population of cells induce SOS very strongly. We further analysed the SOS expression with respect to the time when elongation stopped as a proxy for cell death. For the high-SOS cells, the spike in SOS levels was approximately correlated with cell death, whereas there was no clear correlation between SOS expression and elongation rates for the low-SOS cells (Fig. EV1A,B). Quantification of this observation showed that in all growth conditions, most of the high-SOS cells exhibited a jump in SOS shortly after growth stopped (Fig. EV1C–E). This slight delay is likely due to GFP maturation time. Thus, the high SOS level we observe is likely due to continuous expression from the SOS promoter up until the cells die: the drop in elongation rate (or dilution rate) leads to a smaller volume and, therefore, an increase in GFP concentration.

We then performed the same analysis for the CIP-TET drug combination: overall, more cells survived as expected from the suppressive interaction between CIP and TET. Among cells that died, we still observed two SOS sub-populations (Appendix Fig. S11). However, the proportion of low-SOS and high-SOS cells was very different from the CIP treatment alone, suggesting these two sub-populations do not react similarly to the CIP-TET combination. Indeed, 52% of the total population died and had a low level of SOS under CIP exposure in glu-aa, but these cells represented only 15% of the population in CIP-TET (Fig. 4J). Similarly, in the other growth conditions, the proportion of cells that died and were low-SOS was reduced under CIP-TET (Fig. 4K,L). This suggests that exposure to the combination of antibiotics increases the survival of the cells that do not reach a high-SOS level (Fig. 4J–L, right column magenta). In contrast, the high-SOS sub-population that died was comparatively much less reduced under the drug combination (Fig. 4J–L, yellow), going from 12% to 8% in glu-aa, 25% to 20% in glu, and 30% to 22% in

gly. Taken together, these results suggest that there are likely two different responses to DNA damage among cells that die: one that reaches moderate levels of SOS expression and is affected by TET treatment and a second one which reaches much higher levels of SOS and is hardly sensitive to TET exposure.

## The CIP-TET combination improves the survival of low-SOS cells in a growth-dependent manner

To quantify the importance of the two sub-populations to overall survival, we measured the survival probability of the population in each growth medium separating the contribution of the low and high-SOS cells (Fig. 5A–F). A significant improvement in cell survival under CIP-TET (compared to CIP treatment alone) was observed for both low and high-SOS cells in all growth conditions (Log-rank test, $p < 0.005$). Using the survival fractions for low-SOS cells at the end of the experiment (Fig. 5G), we observed a significant increase under the drug combination compared to the Bliss independence expectation (one-way $t$-test, ANOVA model, $p < 0.005$), indicating a suppressive interaction in all conditions. However, we did not observe a significant suppressive interaction compared to Bliss independence for the high-SOS cells, except for the gly medium (Fig. 5H). This suggests that the drug combination is not, or only slightly, antagonistic for the high-SOS cells.

We noticed that the relative increase in survival appeared to be higher in faster growth conditions than in slow growth conditions for the low-SOS cells. To compare the level of suppression in each growth condition for these cells, we calculated the relative increase in survival under CIP-TET compared to CIP only as $\Sigma = (Sf_{CIP\text{-}TET} - Sf_{CIP})/Sf_{CIP}$, similarly to what has been proposed by Bollenbach et al (2009). The level of suppression was higher in glu-aa (fast growth condition) than in gly—an increase in $\Sigma$ from

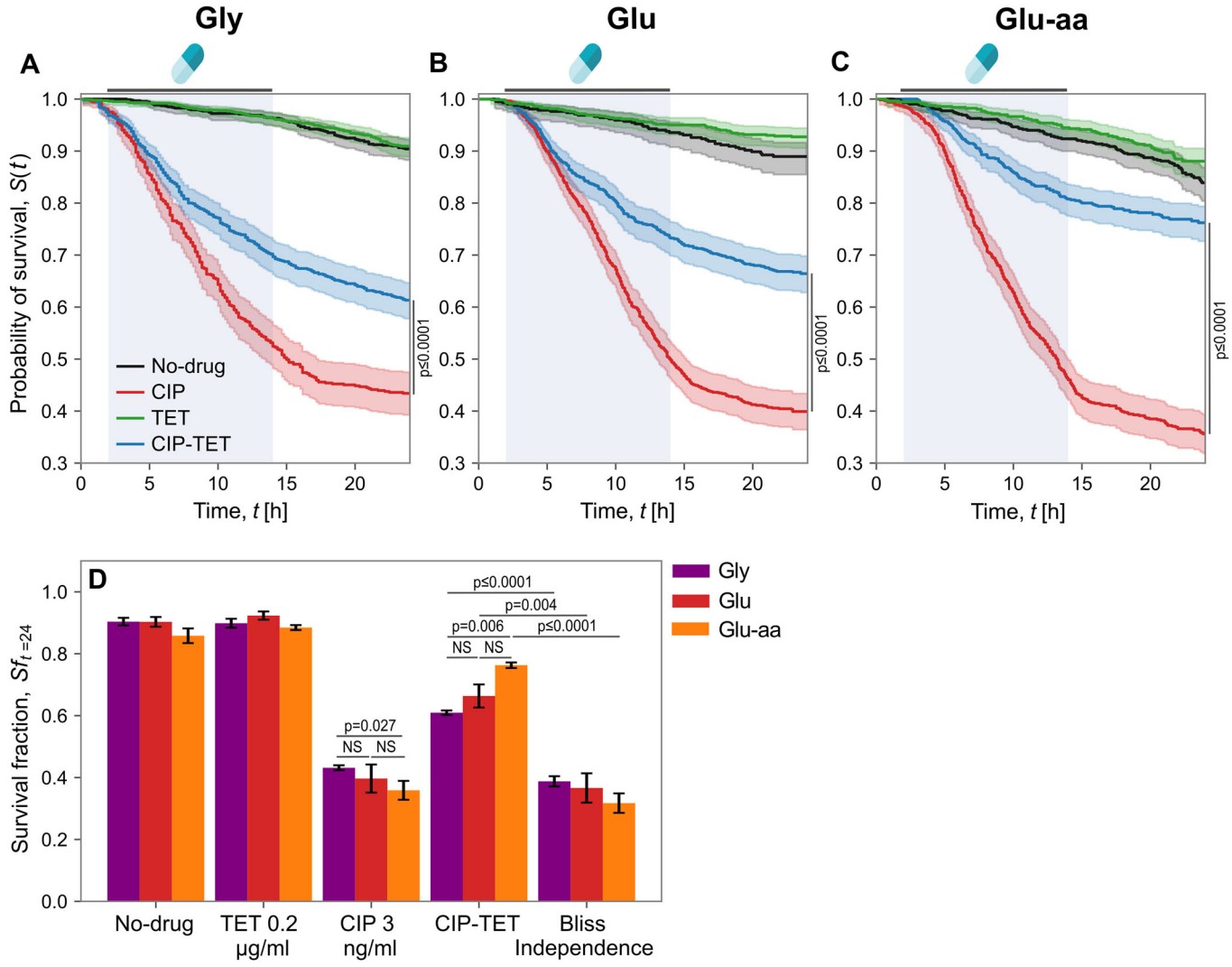

**Figure 3.  The CIP-TET combination improves survival in all growth conditions.**

(A–C) Survival probability curves under different antibiotic treatments in three different growth media. Survival probabilities ($S_t$) were calculated using the Kaplan–Meier estimator and mother cell survival times pooled from at least two experimental replicates consisting of 420–1005 mother cell lineages (per survival curve). Antibiotics were introduced between hours 2–14 (blue shaded area). Shaded area represents 95% confidence intervals of the estimated survival probability. Survival curves for CIP and CIP-TET are significantly different in all growth media (Log-rank test, $p < 0.001$). Survival curves for no-drug control and TET are not significantly different. (D) Mean survival fractions ($Sf$) for all cells quantified at the end of the experiment under antibiotic mono-exposure and drug combination in three different growth media. See 'Methods' for calculation of Bliss independence and statistical analysis. Significant difference of $Sf$ means among and between the CIP and CIP-TET groups were calculated using a two-sided $t$-test. Difference in $Sf$ means between Bliss and CIP-TET were calculated using an ANOVA model (one-sided $t$-test) described in the 'Methods'. All mean survival fractions under CIP-TET were significantly higher than under CIP alone (two-sided $t$-test, $p = 0.002$ for gly, $p = 0.025$ for glu, $p = 0.0002$ for glu-aa). Error bars represent the standard error of the mean. Total number of cells for no-drug control: $n = 868,420,617$; TET: $n = 1005,700,553$; CIP: $n = 558,736,527$; CIP-TET: $n = 756,683,607$ (Key: Gly, Glu, Glu-aa).

0.14 to 0.78 (Figs. 5I and EV2, see 'Discussion'), suggesting that suppression of cell death is growth rate dependent for the low-SOS cells. Taken together, these results suggest that the low-SOS population is protected by treatment with the combination of antibiotics in a growth-dependent manner.

## Discussion

In this work, we quantified the impact of treatment to a sub-lethal concentration of ciprofloxacin in combination with tetracycline at

the single-cell level. We observed that, as expected for a bactericidal antibiotic even at sub-MIC concentration (Coates et al, 2018), treatment with CIP led to significant cell death in a growth-dependent manner when growth is modulated by nutrient quality. The CIP-TET combination markedly increased cell survival, revealing that the improved growth observed at the level of bacterial populations, which underlies the suppressive interaction between CIP and TET (Bollenbach et al, 2009), is due to improved survival. We also observed that suppression was stronger in faster growth conditions due to a combination of two effects: higher frequency and stronger suppression for the

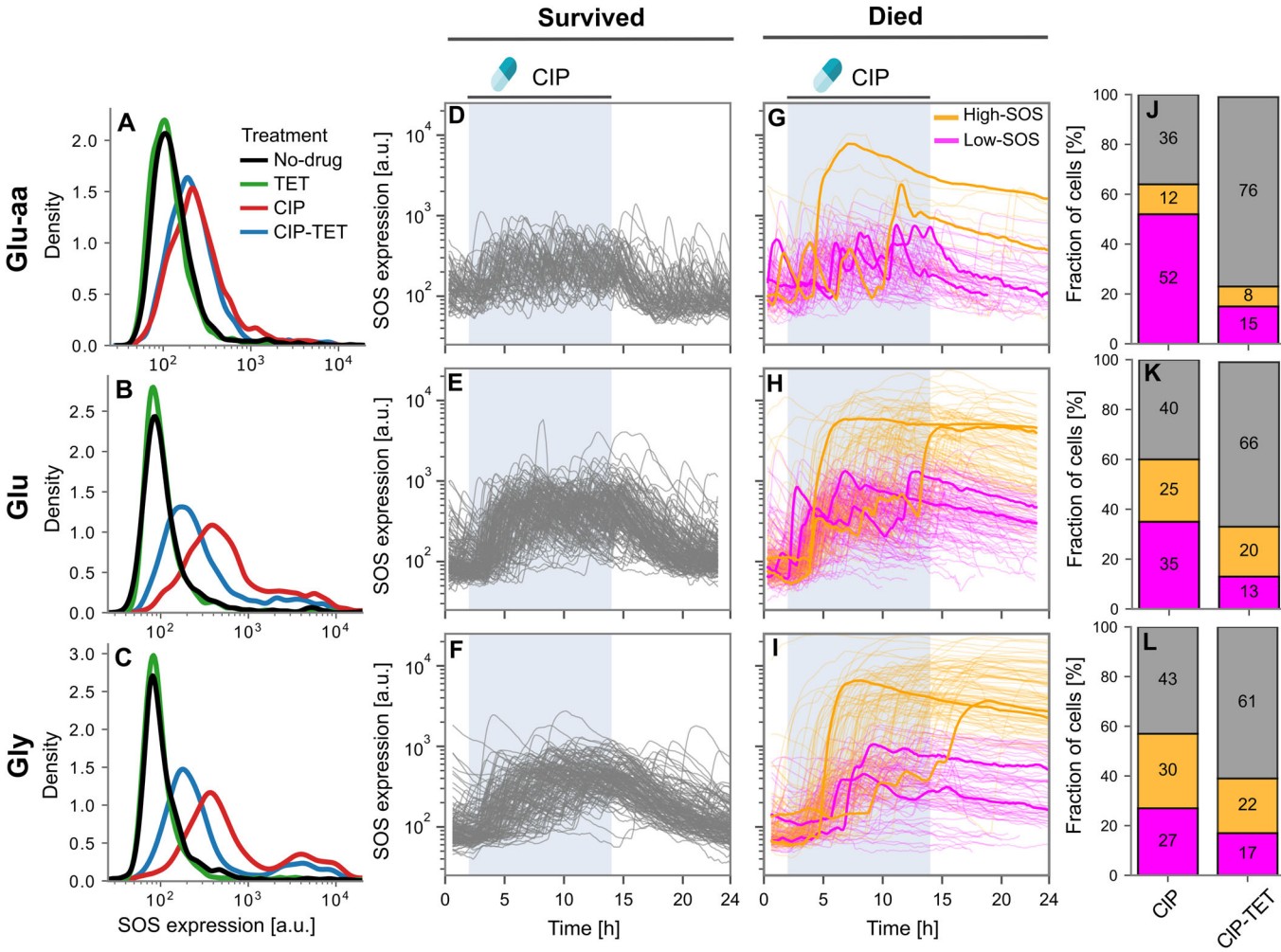

**Figure 4. The SOS response is highly heterogeneous under CIP treatment.**

(A–C) Distributions of single-cell SOS expression from $P_{sulA}$-mGFP for the no-drug control and treatment with TET, CIP and CIP-TET for growth in glu-aa (A), glu (B) and gly (C) medium. Distributions are kernel density estimates of the underlying histogram drawn from the final 2 h of the antibiotic treatment period and pooled from at least two experimental replicates. See Table EV1 for number of cell-cycles making up each distribution. (D–I) Single-cell trajectories of SOS expression classified by cell fate (D–F: cells that survived drug treatment, G–I: cells that died) under CIP treatment in different growth conditions (D,G: glu-aa; E,H: glu; F,I: gly). Dead cells were further classified by expression level (low-SOS: magenta, high-SOS: orange) as described in the Appendix Methods. Example trajectories are highlighted to illustrate the behaviour of the two sub-populations. Shown is data from one experimental replicate for clarity. CIP was introduced between hours 2–14 (blue shaded area). Equivalent figure representing the CIP-TET condition is shown in Appendix Fig. S11. (D): $n = 67$ cells; (E): $n = 140$; (F): $n = 131$; (G): $n = 88$; (H): $n = 221$; (I): $n = 169$. (J–L) Fraction of cells classified as survived (grey), low-SOS dead (magenta), and high-SOS dead (orange) over the entire experiment under CIP and CIP-TET treatment under different growth conditions.

low-SOS cells under CIP-TET in fast growth compared to slow growth.

Indeed, our single-cell results indicate that it is important to distinguish two different sub-populations amongst the bacteria that die under CIP treatment. The first sub-population exhibits a very high level of SOS induction (~10 times more than the rest of the population) at the time of elongation arrest. It is much more abundant in slow growth (glu medium, with a doubling time ≈82 min and gly medium, with doubling time ≈185 min) than in fast growth conditions (glu-aa medium, doubling time ≈40 min) (Table EV1). This might be surprising at first glance, as we recently reported that the rate of high SOS induction in response to a replication-dependant break was higher in fast growth (Jaramillo-

Riveri et al, 2022). However, when treating bacteria with ciprofloxacin, DSBs can happen anywhere on the chromosome and are not necessarily replication-dependent (Zhao et al, 2006), making the consequences of DSB formation very different between the two scenarios. This is because if a DSB occurs in a region of the chromosome that has not been replicated, its repair by homologous recombination would not be possible. Although fast-growing cells have a higher DNA content and thus more potential targets for CIP, they also have higher DNA repair probability due to the presence of multiple chromosome copies. Therefore, we do not expect a high level of irreparable breaks in fast-growing cells for this type of DNA damage. In slow growth conditions, where the cell cycle is longer than 60 min, cells do not undergo multi-fork

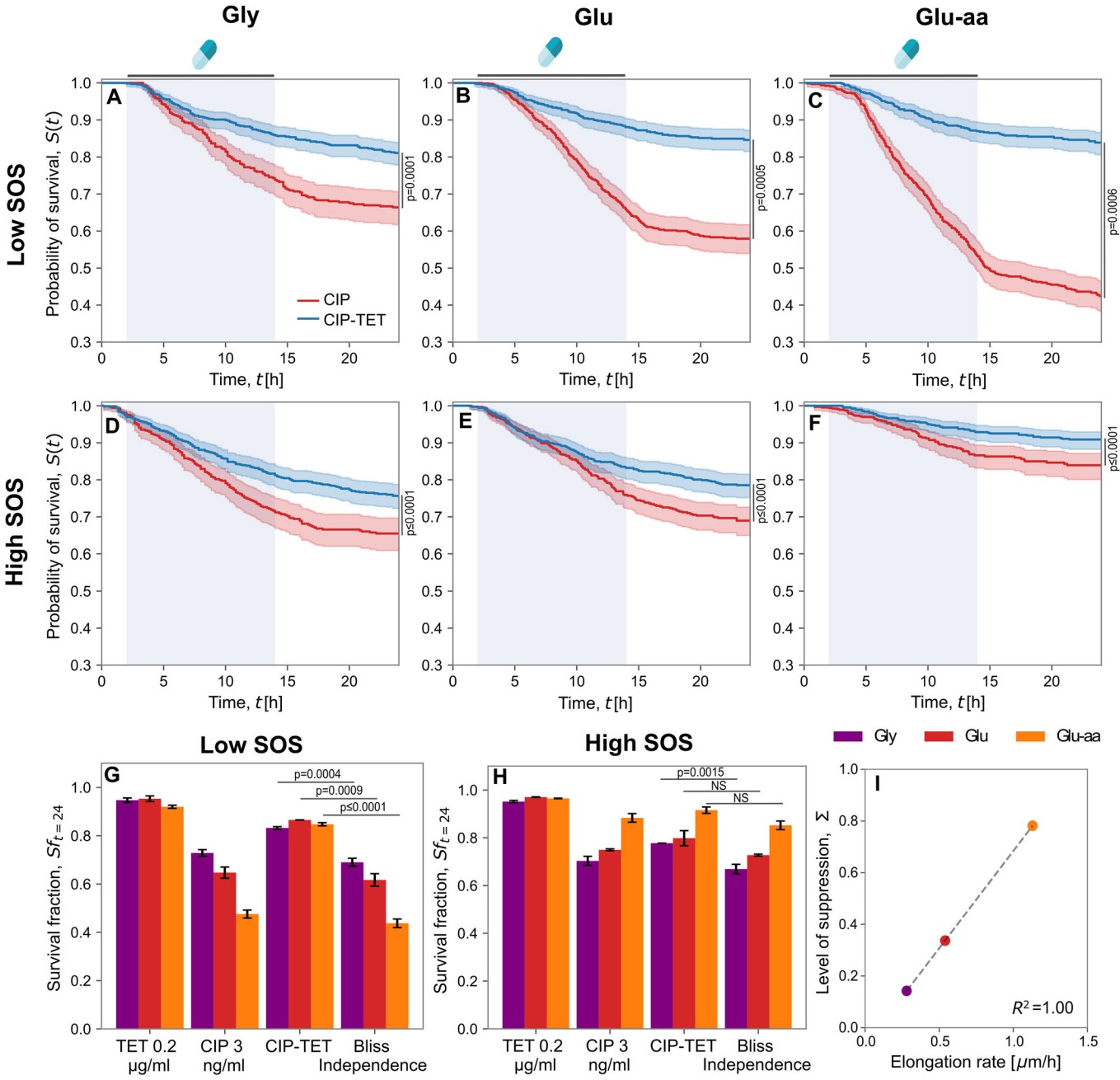

**Figure 5. The CIP-TET combination improves the survival of low-SOS cells in a growth-dependent manner.**

(A–F) Survival probability curves for low-SOS cells (A–C) and high-SOS cells (D–F) under CIP and CIP-TET treatments in three different growth media. Survival probabilities ($S_t$) were calculated using the Kaplan–Meier estimator and mother cell survival times pooled from at least two experimental replicates. Antibiotics were introduced between hours 2–14 (blue shaded area). Shaded area represents 95% confidence intervals of the estimated survival probability. Survival curves under CIP and CIP-TET are significantly different in all growth media (Log-rank test). (G, H) Mean survival fractions ($Sf$) for low-SOS cell death (G) and high-SOS cell death (H) quantified at the end of the experiment under antibiotic mono-exposure and drug combination in three different growth media from at least two experimental replicates. See 'Methods' for calculation of Bliss independence and statistical analysis. Difference in $Sf$ means between Bliss and CIP-TET were calculated using an ANOVA model (one-sided $t$-test) described in the 'Methods'. Error bars represent the standard error of the mean. (I) Level of suppression of low-SOS cell death quantified in three growth conditions plotted against drug-free median single-cell elongation rate (see 'Methods' for calculation).

replication and a larger proportion of the chromosome is unreplicated during the cell cycle (Cooper and Helmstetter, 1968). We, therefore, expect a larger number of irreparable breaks in glu and gly, and it is likely that these breaks lead to high SOS

induction before cell death. In other words, the cells with a high-SOS phenotype likely die because of an irreparable DSB. Treatment with TET, at the low concentration we used, would not change these cells' replication status, therefore explaining the lack of

suppression of the CIP-TET combination in this high-SOS subpopulation.

The second population of cells that died under CIP exposure did not exhibit a level of SOS higher than that of the cells that survived. Nevertheless, they stopped cell elongation and protein production. Contrary to the high-SOS population, these cells were more abundant in fast growth (glu-aa) than in glu and gly, and their frequency diminished markedly when exposed to the CIP-TET antibiotic combination. This indicates that the suppressive effect of CIP-TET is mostly driven by the increased survival of these low-SOS cells. Because of their limited SOS induction and higher abundance in glu-aa (when a large part of the chromosome is present in multiple copies because of multi-fork replication), we hypothesise that cellular death is not due to an irreparable break, but rather that these cells die through another molecular process that has yet to be fully characterized. To better understand their characteristics, we computed the elongation rate over a two-hour period before death of the low-SOS cells, the high-SOS cells, and the cells that survived (for which we used a 2-hour period during antibiotic exposure). In all nutrient conditions, the elongation rate of the low-SOS cells that died was higher on average than that of the cells that survived or induced high SOS levels (Appendix Figs. S12 and S13, left panel). However, this difference disappeared in the CIP-TET condition suggesting that the reduction in elongation rate due to TET may underlie the improved survival of these cells. This effect was stronger in glu-aa than in glu and gly. This is expected because we observed a growth-dependent effect of TET on cell elongation: the elongation rate of individual cells treated with TET decreased to a greater extent in glu-aa than in glu and gly (Table EV1, Figs. 1J, S7F and S8F) similar to what has been documented previously at the population level (Greulich et al, 2015; Scott et al, 2010). These results suggest that the death of the low-SOS cells is linked to a high elongation rate, which is alleviated by TET treatment (see below for discussion of potential mechanisms). Our observation that cells exposed to ciprofloxacin can die through two different phenotypes (high and low SOS expression) is reminiscent of the two sub-phenotypes (high and low TetA expression) observed when resistant *E. coli* cells (*tet* operon) are treated with tetracycline (Schultz et al, 2017). Phenotypic heterogeneity in the regulation of stress responses can result in distinct routes to cell death that can only be revealed using single-cell methods.

While our observations help understand the growth dependence of the suppressive effect of TET on the low-SOS cells, they do not fully elucidate the molecular mechanisms that lead to the death of these cells. Recent work suggests that cells that elongate transiently more slowly and induce increased levels of stress responses, such as acid resistance, survive better exposure to high levels of ciprofloxacin (Sampaio et al, 2022). However, we cannot directly link this phenomenon to our observations: in our case, elongation reduction by TET treatment is expected to also lead to a reduction of protein production and, therefore, limit the induction of various stress responses. Indeed, we observe that the increase in survival in CIP-TET for the low-SOS cells is not due to an increase in SOS expression; as expected from proteome global constraints, the SOS response is reduced in gly and glu under CIP-TET compared to CIP treatment and does not change in glu-aa (Fig. 4A–C). As shown in Fig. 5 and Fig. EV2, we have observed that the fraction of low-SOS cells that eventually die under CIP correlates positively with elongation rate. This is reminiscent of the higher mass fraction of ribosomes in fast growth predicted by the growth laws (Scott et al, 2010); it is tempting to speculate that these cells die because of toxicity due to an irreversible impairment of their ribosomes, which would be more frequent when ribosomes are more abundant. For example, the YafO toxin (Singletary et al, 2009; Zhang et al, 2009), which is SOS-regulated, has been reported to target the ribosomes and could, at least in part, lead to this toxicity. Tetracycline, which also binds the ribosomes, could act as a competitive inhibitor of the toxins thus preventing cell death. Alternatively, it has been proposed that cell death under CIP exposure could be linked to the formation of reactive oxygen species (Hong et al, 2020; Kohanski et al, 2007) or to an imbalance in energy metabolism where ATP becomes limiting for DNA repair (Chevereau and Bollenbach, 2015). Treatment with TET, which limits the high-energy-consuming protein synthesis machinery, could free up ATP for DNA repair. It has also been shown that bacteriostatic antibiotics such as TET reduce cellular respiration, which results in decreased efficacy of bactericidal antibiotics such as CIP (Lobritz et al, 2015). Our results suggest that full elucidation of these mechanisms requires detailed quantification at the single-cell level using, for example, single-cell reporters of ATP concentration as has been done recently (Lin and Jacobs-Wagner, 2022; Yaginuma et al, 2014). In the longer term, a quantitative understanding of how individual cells respond to treatment can inform emergent population-level effects and improve the modelling of bacterial physiological responses to drug combinations.

## Methods

**Reagents and tools table**

| Reagent/Resource | Reference or Source | Identifier or Catalog Number |
|---|---|---|
| **Experimental models** | | |
| *Escherichia coli* K-12 MG1655 | CGSC 7740 | |
| **Recombinant DNA** | | |
| pE-FLP | St.-Pierre et al (2013) | Appendix Table S3 |
| **Oligonucleotides and other sequence-based reagents** | | |
| PCR primers | This study | Appendix Table S4 |
| **Chemicals** | | |
| Ciprofloxacin hydrochloride | APExBIO | C5539 |
| Tetracycline hydrochloride | Fisher BioReagents | BP912-100 |
| MEM 50X essential amino acids | Gibco | 11130051 |
| MEM 100X non-essential amino acids | Gibco | 11140035 |
| D(+)-Glucose anhydrous | Scientific Laboratory Supplies | CHE1806 |
| Glycerol | Sigma-Aldrich | G9012 |

| Reagent/Resource | Reference or Source | Identifier or Catalog Number |
|---|---|---|
| Tween-20 Surfact-Amps detergent solution | Thermo Scientific | 85113 |
| Poly-dimethylsiloxane (PDMS) | Sylgard, Dow Corning | Silicone Elastomer Kit 184 |
| **Software** | | |
| Fiji | https://imagej.net/software/fiji/ | |
| BACMMAN | https://github.com/jeanollion/bacmman Ollion et al (2019) | |
| Python | https://www.python.org | |
| Matlab | https://uk.mathworks.com/ | |
| **Other** | | |
| FLUOstar Omega microplate reader | BMG Labtech | |
| Flat-bottomed 96-well plates | Corning Costar | |
| 21-guage blunt needle | OctoInkjet | |
| Biopsy punch ID 0.5 mm | World Precision Instruments Ltd | |
| Automatic bioreactor | Ogibiotec | |
| Coverslip, 24 × 60 mm, #1.5 | Duran | |
| Peristaltic pump | Ismatec | IPC ISM932D |
| DAQ | Measurement Computing | USB-1408FS |

## Culture conditions

For all microscopy and batch experiments, cell cultures were grown in M9-based media. The composition of the M9 salts was as follows: 49 mM $Na_2HPO_4$, 22 mM $KH_2PO_4$, 8.6 mM NaCl, 19 mM $NH_4Cl$, 2 mM $MgSO_4$, and 0.1 mM $CaCl_2$ (adjusted to pH = 7). This was supplemented with either 0.5% v/v glycerol (Sigma-Aldrich, G9012), giving rise to the "gly" medium; 0.5% w/v glucose (D(+)-glucose anhydrous, Scientific Laboratory Supplies, CHE1806), "glu" medium; in the third medium, "glu-aa", a mix of amino acids (1× MEM Non-Essential Amino Acids and 1× MEM Essential Amino Acids, both manufactured by Gibco) was added with the glucose. For strains and plasmid construction, cells were grown in LB, or LB agar supplemented with the corresponding selection marker (kanamycin 50 µg/mL). All cultures were grown at 37 °C (150 rpm agitation) in 50 mL falcon tubes with no more than 5 mL of liquid volume or 500 mL Erlenmeyer flasks with no more than 50 mL of liquid volume, unless otherwise stated.

Antibiotics used in this study are listed as follows: ciprofloxacin hydrochloride (APExBIO, C5539) and tetracycline hydrochloride (Fisher BioReagents, BP912-100). Antibiotic stocks were made up in sterile distilled water and filter-sterilised (0.22 µm). Stock concentrations were 5 mg/mL for ciprofloxacin (pH-adjusted to 4.62) and 10 mg/mL for tetracycline. Antibiotic stocks were aliquoted into small volumes (<200 µL) and frozen at −20 °C. For each experiment, new antibiotic stocks were thawed on the same day and never re-used. Working stocks were diluted using sterile M9-based medium. Fresh tetracycline stocks were made up annually using a new powder batch due to its low long-term stability. When used, antibiotics were always shielded from light.

## Strain construction

*Escherichia coli* K-12 MG1655 was used as WT strain in this study. Gene expression reporters (SOS expression: $P_{sulA}$-*mGFP*; constitutive expression: $P_{tet01}$-*mKate2*) were inserted into the genome by clone-integration as described previously (Jaramillo-Riveri et al, 2022). For the *motA* gene deletion strain, a single gene deletion knockout (from the Keio deletion collection (Baba et al, 2006)) was introduced in our strain using P1 transduction. The *motA* gene was removed to inhibit flagellum activity to improve cell retention in the microfluidic microchannels. The antibiotic resistance selection markers were removed by transformation of the pE-FLP plasmid (St-Pierre et al, 2013). After construction, inserts were checked by PCR amplification and sequencing of the modified chromosomal region. The strains, plasmids, and primers used are listed in Appendix Tables S2, S3, and S4, respectively.

## Fluorescence microscopy

All images were captured using a Nikon Ti-E inverted microscope equipped with EMCCD Camera (iXon Ultra 897, Andor), a SpectraX Line engine (Lumencor) and a 100× Nikon TIRF objective (NA 1.49, oil immersion). Nikon Perfect-Focus system was used for continuous maintenance of focus. The filter set for imaging mGFP consisted of ET480/40× (excitation), T510LPXR (dichroic), and ET535/50m (emission); whereas for mKate2 the set ET572/35× (excitation), T590LPXR (dichroic), and ET632/60m (emission) was used. Filters used were purchased from Chroma. mGFP fluorescence was measured using 80 ms exposure, whereas mKate fluorescence was imaged for 100 ms, both at minimal camera gain and maximum lamp intensity. Brightfield images were also acquired at each frame using a 20 ms exposure. The microscope and temperature chamber was turned on to 37 °C at least 2 h before imaging. The microscope was controlled from MATLAB via MicroManager (Edelstein et al, 2014) using a custom-made user interface developed previously (Jaramillo-Riveri, 2019). The code is accessible at https://gitlab.com/MEKlab/MicroscopeControl. Images were saved in *tiff* format as one file per fluorescence channel per frame.

## Mother machine experiments

### Microfluidic device design and fabrication
The design and fabrication of the mother machine device was described previously (Jaramillo-Riveri et al, 2022). The silicone wafers were manufactured by Micro Resist Technology GmbH. Microchannel dimensions were optimised for each medium and are indicated in Appendix Table S5.

### Culture preparation and experimental design
Cells were inoculated in 5 mL LB from a −80 °C stock and grown overnight. The next morning (day of experiment), cells were subcultured (1:1000 dilution factor) into M9-based medium and incubated at 37 °C with shaking (150 rpm). When the culture

reached an $OD_{600}$ of 0.2, cells were harvested for inoculation in the mother machine. For the slow growth medium (gly), an extra step was added: After 1:1000 dilution from the LB overnight into M9-based medium (late afternoon), the culture was incubated overnight. The next morning, cells were diluted again (1:200) and allowed to grow until $OD_{600} = 0.2$ before being harvested for the mother machine that same day.

Once the culture was ready for inoculation, Tween-20 Surfact-Amps detergent solution (Thermo Scientific, 85113) was added to prevent cell clumping (0.01% final concentration) before being concentrated 100-fold by centrifugation (4000 rpm for 5 min). Before sample loading, the chip was passivated with Tween-20 (0.01% final concentration) for at least 1 h. The concentrated cell culture was briefly vortexed and then injected into the inlets of the microfluidic chip using a 1 mL syringe with a 21-gauge blunt needle (OctoInkjet). The channel inlets were sealed temporarily using adhesive tape and left at 37 °C for ~30 min to allow cells to diffuse into the microchannels. To further assist cell loading, the cells were then spun into the microchannels by centrifugation at 4000 rpm ($3220 \times g$) for 5 min using a custom-built mount. Centrifugation was essential for loading of non-motile cells. The chip was then mounted on the microscope using a custom-built mount and the loading efficiency was evaluated. Before and after each experiment, the inlet and outlet tubing (0.44 mm ID) were sterilised with 1% bleach (equivalent to 10,000 ppm chlorine), followed by 10% ethanol, rinsed with Milli-Q water, and then primed with fresh medium. Tween-20 (0.01% final concentration) was also added to the medium reservoir to prevent cell adhesion and biofilm formation within the tubing and device. Inlet and outlet tubing was then connected to the inlet and outlet channels of the microfluidic device via 21-gauge blunt needles with the medium reservoir on one end and effluent reservoir on the other end. Tubing was connected to the reservoirs via a custom-built manifold. Inlet tubing was also connected to three-way valves (two inlets, one outlet) which were used to switch between no-drug and antibiotic influent reservoirs. Valves were controlled automatically via MATLAB communication with a data acquisition device (DAQ, Measurement Computing, USB-1408FS). Valve-outlet tubing was then connected to a peristaltic pump (Ismatec IPC ISM932D) which 'pushed' (positive-pressure flow is advantageous as it avoids feature collapse due to negative-pressure driven flow) medium through the microfluidic chip into the effluent flask.

Once cells were loaded into the microfluidic device, medium flow was used to flush cells out the main trench using a peristaltic pump at a flow rate of 1.5–2 mL/h and then lowered to 1 mL/h for the duration of the experiment. All flow channels received fresh medium without antibiotic for at least 2 h to allow for recovery. Images were then acquired at 5 min intervals for glu-aa and glu media, and 10 min for gly medium. When imaging started, all channels received medium without antibiotic for a further 6 h, followed by 12 h of antibiotic exposure, and then switched back to medium without antibiotic for at least 10 h, unless otherwise stated. The pre-antibiotic period ensures all cells have acclimated to the system and are undergoing balanced exponential growth. The post-antibiotic period (or recovery period) allows the resuscitation of persister cells, if present. Antibiotics solutions were prepared on the day of the experiment to the desired concentration in M9-based medium in separate 50 mL falcon tubes. These were kept at 37 °C for the duration of the experiment and wrapped in foil to protect

from light. The microfluidic device has four independent flow channels. Thus, for each experiment, four treatments ran simultaneously: A no-drug control, CIP 3 ng/mL, TET 0.2 µg/mL, and the drug combination (CIP-TET) at the same concentrations. At least two experimental replicates were performed on different days, except for CIP in glu-aa, the control in gly, and TET in gly, where three replicates were performed. No blinding was done in this study.

### Image analysis

Cell segmentation and lineage tracking were performed using BACMMAN run in Fiji (Ollion et al, 2019). The BACMMAN configuration was adapted to segment and track cells based on fluorescence images. Images (*tiff* format) were imported in BACMMAN and pre-processed prior to segmentation. This included image rotation to ensure channels were vertically oriented and the channel opening was at the bottom, and cropping of images to include only the area consisting of microchannels. Background subtraction was performed for both fluorescent channels using ImageJ's subtract background algorithm (rolling ball method, radius = 8). Joint segmentation and tracking of cells were performed using a DistNet model trained on our datasets. DistNet is based on a deep neural network (Ollion and Ollion, 2020). Segmentation was performed on the mKate fluorescence channel. Segmentation parameters were optimised for each dataset. Curation of segmentation and tracking was carried out using BACMMAN's interactive graphical interface. Although automated segmentation and tracking was mostly accurate, occasionally errors were produced. Thus, almost every lineage was manually checked, with CIP exposure datasets requiring the most curation mainly due to excessive elongation of cells. At this point obvious irregularities were removed including mother cells that did not grow for the duration of the experiment and those that were already excessively elongated at the beginning of the experiment. Positions with channel deformities and where loss of focus occurred were also discarded.

Single-cell fluorescent intensities were calculated by dividing the sum of pixel intensity values by the cell area. Cell length was determined as the maximal distance between two points of the cell contour (FeretMax). Elongation rates were calculated per cell-cycle (generation) by fitting an exponential function, $S_t = S_0 e^{\mu t}$, on the estimated length of cells, where $\mu$ is the specific growth rate constant (in $h^{-1}$) henceforth referred to as elongation rate, $S_0$ is the length at birth, $S_t$ is the length at time $t$. A minimum of 3 frames per generation was imposed as a fitting constraint. Cell fluorescence and morphology measurements were then exported from BACMMAN into csv files for further analysis.

### Data analysis

Data analysis and plotting was conducted using Python in Jupyter notebooks. The first 4 h of all experiments were discarded to ensure only cells at steady-state were analysed. Thus, each dataset begins with 2 h of no-drug medium before antibiotic exposure. For antibiotic treatment experiments, we removed lineages that were initially filamented or non-growing in the first 2 h. Some cells that filamented extended beyond the length of the microchannels (classified as hyper-filamented cells). We only tracked a cell that filamented up until it reached the maximum microchannel length since cell length measurements beyond that are invalid.

Classification of cell fate (illustrated in Appendix Fig. S9) and sub-classification of low & high-SOS inducers (threshold shown in Appendix Table S6) is described in the Appendix Methods. Survival analysis was carried out using a Kaplan–Meier estimator (Kaplan and Meier, 1958) described in detail in the Appendix Methods.

The expected survival fractions (*Sf*s) under the antibiotic combination can be calculated according to the Bliss definition of independent drug interactions as the product or, equivalently, as the log-transformed sum of the individual drug effects (Bliss, 1939; Demidenko and Miller, 2019). The latter method simplifies statistical testing by turning the Bliss model into a linear equation (Demidenko and Miller, 2019). The Bliss independence expected *Sf* presented in Figs. 3D and 5G,H was derived from the log-transformed pairwise sums of the replicate *Sf*s in each group: $Bliss_{ij}$ = $\exp(\ln(Sf_{CIP_{ij}}) + \ln(Sf_{TET_{ij}}))$, where *i* = 1, 2, 3 for the three growth media and *j* = 1,..., $n_i$, where $n_i$ is the number of replicates in the *i*th group (Demidenko and Miller, 2019). The corresponding error bars (standard error of the mean) were calculated as: $SEM_{CIP-TET} = \sqrt{\frac{\sigma_{CIP}}{\sqrt{n_{CIP}}} + \frac{\sigma_{TET}}{\sqrt{n_{TET}}}}$, where $\sigma$ is the standard deviation of the respective single-drug $\overline{Sf}$s and *n* is the number of experimental replicates. The level of suppression calculated in Fig. 5I was defined as the relative change in the mean survival fraction $\Sigma = (\overline{Sf}_{CIP-TET} - \overline{Sf}_{CIP})/\overline{Sf}_{CIP}$, where $\overline{Sf}_{CIP-TET}$ is the mean survival fraction under the drug combination and $\overline{Sf}_{CIP}$ is the mean survival fraction under CIP which is antagonised by TET. Equation adapted from Bollenbach et al (2009).

## Statistical analysis

Statistical analysis of the Kaplan–Meier survival curves were conducted using the Log-rank test. For the survival fractions reported at the end of the experiment, a two-sided *t*-test was used to test for significant differences in means between antibiotic treatments and growth media. To test the null hypothesis of drug independence according to Bliss ($H_0$: $\ln(Sf_{CIP})$ + $\ln(Sf_{TET})$ − $\ln(Sf_{CIP-TET})$ = 0), we used an ANOVA model described by Demidenko and Miller (2019) (see Appendix Methods for full description of model). We computed the one-sided *p*-value to determine if the observed *Sf* under CIP-TET was significantly higher than the expected *Sf* according to Bliss independence. All statistical analysis was performed in Python.

## Data availability

The datasets and computer code produced in this study are available in the following databases: Mother machine image data: BioImage Archive S-BIAD1260 (https://www.ebi.ac.uk/biostudies/bioimages/studies/S-BIAD1260). Bulk growth rate data: Zenodo 15311905 (https://zenodo.org/records/15311905). Python code used to process the data and generate the plots: GitLab (https://gitlab.com/MEKlab/single-cell-suppression-2024).

The source data of this paper are collected in the following database record: biostudies:S-SCDT-10_1038-S44320-025-00162-w.

## Peer review information

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

## Acknowledgements

We wish to thank Matthew Scott for helpful discussions and Louise Goossens for performing experiments related to Appendix Fig. S3. This work was supported by a Wellcome Trust Investigator Award (Grant No. 205008/Z/16/Z) to Meriem El Karoui, a Darwin Trust of Edinburgh PhD studentship to James Broughton, and by the United Kingdom Research and Innovation (grant EP/

S02431X/1), UKRI Centre for Doctoral Training in Biomedical AI at the University of Edinburgh, School of Informatics (PhD studentship to Achille Fraisse). For the purpose of open access, the authors have applied a Creative Commons attribution (CC BY) licence to any author-accepted manuscript version arising.

## Author contributions

**James Broughton**: Conceptualization; Data curation; Formal analysis; Investigation; Visualization; Methodology; Writing—original draft; Writing—review and editing. **Achille Fraisse**: Formal analysis; Visualization; Writing—review and editing. **Meriem El Karoui**: Conceptualization; Supervision; Funding acquisition; Writing—original draft; Project administration; Writing—review and editing.

Source data underlying figure panels in this paper may have individual authorship assigned. Where available, figure panel/source data authorship is listed in the following database record: biostudies:S-SCDT-10_1038-S44320-025-00162-w.

## Disclosure and competing interests statement

The authors declare no competing interests.

# Expanded View Figures

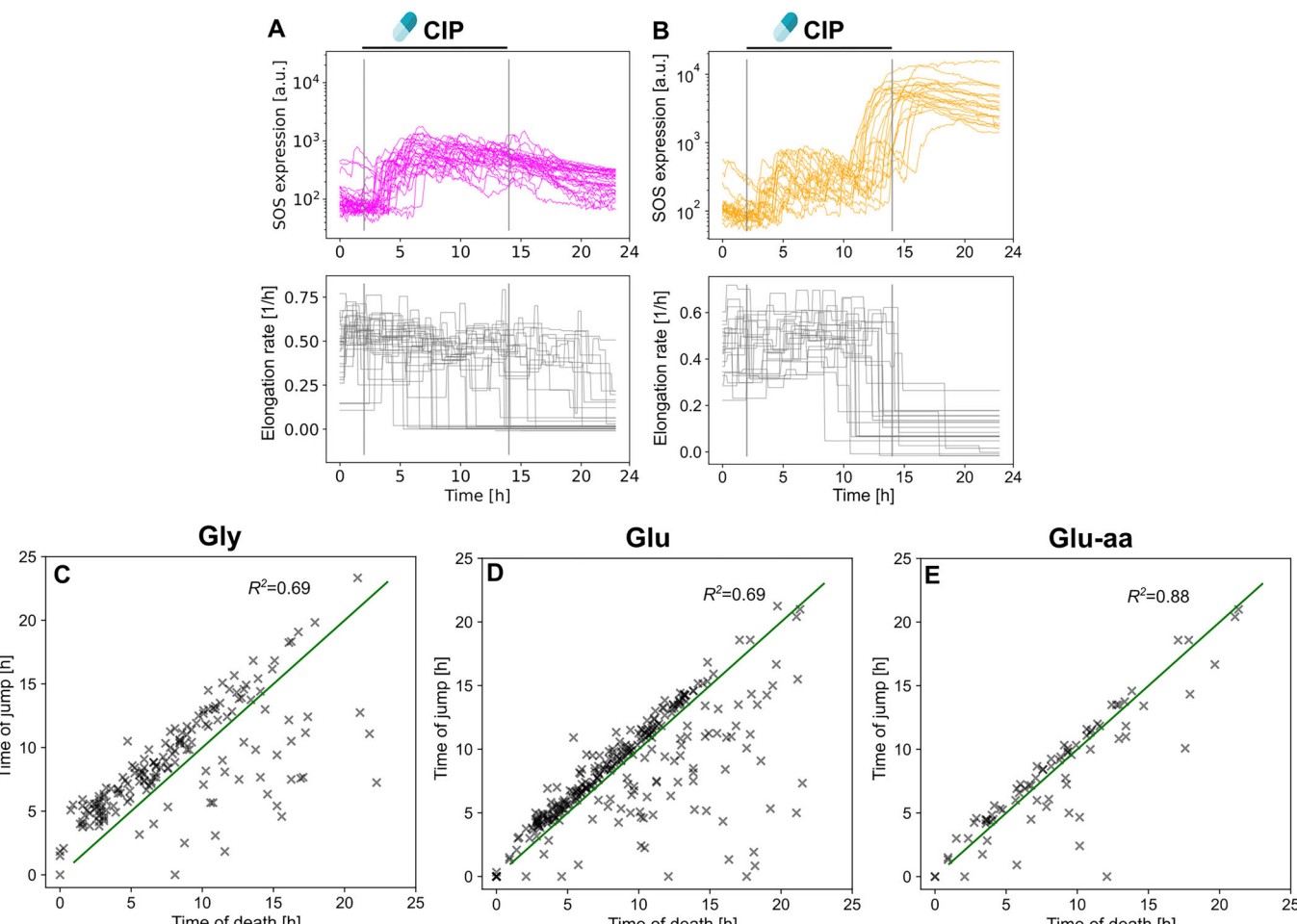

**Figure EV1.  High-SOS stress jump correlates with the time of elongation arrest.**

(**A**) Single-cell trajectories of SOS expression and elongation rate for low-SOS cells that died under CIP treatment for growth in the Glu medium. (**B**) Single-cell trajectories of SOS expression and elongation rate for a representative group of high-SOS cells that died at similar times under CIP treatment for growth in the glu medium. We noticed that elongation arrest correlated with the time of jump in SOS levels. One group of cells with similar times of death from one experiment were identified and plotted in (**B**). CIP was introduced between hours 2–14 (vertical lines). (**C–E**) Time of death (defined by the time of elongation arrest) plotted against the time of jump (determined by the time the SOS expression crosses 1300 a.u.) for individual high-SOS cells (black crosses) from CIP treatment experiments. The green line shows the x = y function and $R^2$ is the correlation coefficient.

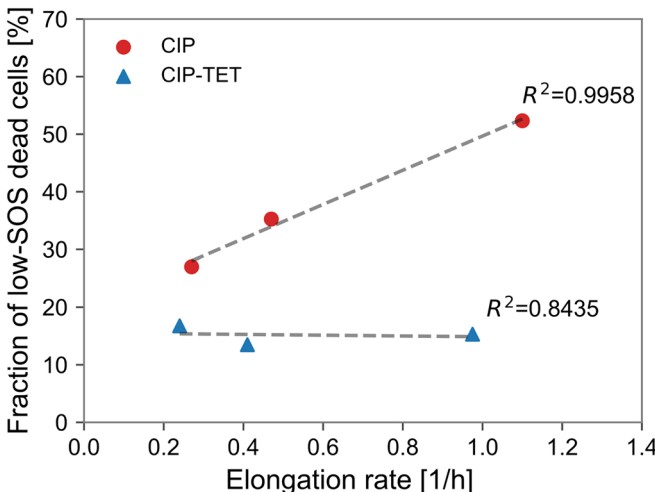

**Figure EV2.  Tetracycline eliminates growth-dependence in low-SOS dead cells under ciprofloxacin.**

Fraction of low-SOS cells that died under CIP and CIP-TET treatment in different growth conditions. Fractions are plotted against the median single-cell elongation rate of surviving cells calculated from the final 2 h of antibiotic treatment. $R^2$ is the correlation coefficient.

  