## [Peer Review File · Molecular Systems Biology]

Suppression of bacterial cell death underlies the antagonism between ciprofloxacin and tetracycline

James Broughton, Achille Fraisse, and Meriem El Karoui

Corresponding author(s): Meriem El Karoui (meriem.elkaroui@ed.ac.uk) , Meriem El Karoui (meriem.elkaroui@ed.ac.uk), James Broughton (j.broughton@ed.ac.uk)

Review Timeline:

Submission Date:	7th Aug 24
Editorial Decision:	30th Sep 24
Revision Received:	6th Jun 25
Editorial Decision:	4th Jul 25
Revision Received:	12th Aug 25
Accepted:	3rd Sep 25

Editor: Poonam Bheda

Transaction Report:

30th Sep 2024

Manuscript Number: MSB-2024-12561

Title: Suppression of bacterial cell death underlies the antagonism between ciprofloxacin and tetracycline

Dear Prof El Karoui,

Thank you for the submission of your manuscript to Molecular Systems Biology. We have now received feedback from the two reviewers who agreed to evaluate your manuscript. As you will see from the reports below, the referees acknowledge the interest of the study and are overall supporting publication of your work pending appropriate revisions.

I think that the recommendations of the reviewers are rather clear and I therefore do not see the need to repeat the comments listed below. One of the important suggestions would be to place your findings in the context of the existing literature. All other issues raised would need to be satisfactorily addressed. Please let me know in case you would like to discuss in further detail any of the comments, I would be happy to schedule a call.

We require:

1) A .docx formatted version of the manuscript text (including legends for main figures, EV figures and tables). Please make sure that the changes are highlighted to be clearly visible. Alternatively you may choose to submit your manuscript as a LaTeX file.

4) A .docx formatted letter INCLUDING the reviewers' reports and your detailed point-by-point responses to their comments. As part of the EMBO Press transparent editorial process, the point-by-point response is part of the Peer Review File (PRF), which will be published alongside your paper.

5) A complete author checklist, which you can download from our author guidelines (<https://www.embopress.org/page/journal/17574684/authorguide#submissionofrevisions>). Please insert information in the checklist that is also reflected in the manuscript. The completed author checklist will also be part of the PRF.

6) Please note that all corresponding authors are required to supply an ORCID ID for their name upon submission of a revised manuscript.

7) It is mandatory to include a 'Data Availability' section after the Materials and Methods. Before submitting your revision, primary datasets produced in this study need to be deposited in an appropriate public database, and the accession numbers and database listed under 'Data Availability'. Please remember to provide a reviewer password if the datasets are not yet public (see <https://www.embopress.org/page/journal/17574684/authorguide#dataavailability>).

This study includes no data deposited in external repositories.

8) All Materials and Methods need to be described in the main text using our 'Structured Methods' format, which is required for all research articles. According to this format, the Methods section includes a Reagents and Tools Table (listing key reagents, experimental models, software and relevant equipment and including their sources and relevant identifiers) followed by a Methods and Protocols section describing the methods using a step-by-step protocol format. The aim is to facilitate adoption of the methodologies across labs. Please upload the Reagents and Tools table as a separate document when submitting your revised manuscript. More information on how to adhere to this format as well as a downloadable template (.docx) for the Reagents and Tools Table can be found in our author guidelines:

<https://www.embopress.org/page/journal/17444292/authorguide#structuredmethods>

An example of a Method paper with Structured Methods can be found here:
<https://www.embopress.org/doi/10.15252/msb.20178071>.

9) For data quantification: please specify the name of the statistical test used to generate error bars and P values, the number (n) of independent experiments (specify technical or biological replicates) underlying each data point and the test used to calculate p-values in each figure legend. The figure legends should contain a basic description of n, P and the test applied. Graphs must include a description of the bars and the error bars (s.d., s.e.m.). Please provide exact p values.

10) Our journal encourages inclusion of *data citations in the reference list* to directly cite datasets that were re-used and obtained from public databases. Data citations in the article text are distinct from normal bibliographical citations and should directly link to the database records from which the data can be accessed. In the main text, data citations are formatted as follows: "Data ref: Smith et al, 2001" or "Data ref: NCBI Sequence Read Archive PRJNA342805, 2017". In the Reference list, data citations must be labeled with "[DATASET]". A data reference must provide the database name, accession number/identifiers and a resolvable link to the landing page from which the data can be accessed at the end of the reference. Further instructions are available at .

11) We replaced Supplementary Information with Expanded View (EV) Figures and Tables that are collapsible/expandable online. A maximum of 5 EV Figures can be typeset. EV Figures should be cited as 'Figure EV1, Figure EV2' etc... in the text and their respective legends should be included in the main text after the legends of regular figures.

<https://www.embopress.org/page/journal/17574684/authorguide#expandedview>

13) Author contributions: CRediT has replaced the traditional author contributions section because it offers a systematic machine readable author contributions format that allows for more effective research assessment. Please remove the Authors Contributions from the manuscript and use the free text boxes beneath each contributing author's name in our system to add specific details on the author's contribution. More information is available in our guide to authors.

14) Disclosure statement and competing interests: We updated our journal's competing interests policy in January 2022 and request authors to consider both actual and perceived competing interests. Please review the policy

<https://www.embopress.org/competing-interests> and update your competing interests if necessary.

Please also suggest a striking image or visual abstract to illustrate your article as a PNG file 550 px wide x 300-600 px high. Share synopsis text and image, as well as eTOC:

Please note that these would be the final versions and changes during proofing are usually not allowed

16) As part of the EMBO Publications transparent editorial process initiative (see our policy here:

https://www.embopress.org/transparent-process#Review_Process), Molecular Systems Biology will publish online a Peer Review File (PRF) to accompany accepted manuscripts.

In the event of acceptance, this file will be published in conjunction with your paper and will include the anonymous referee reports, your point-by-point response and all pertinent correspondence relating to the manuscript. Let us know whether you agree with the publication of the PRF and as here, if you want to remove or not any figures from it prior to publication.

Please note that the Authors checklist will be published at the end of the PRF.

Molecular Systems Biology has a "scooping protection" policy, whereby similar findings that are published by others during review or revision are not a criterion for rejection. Should you decide to submit a revised version, I do ask that you get in touch after three months if you have not completed it, to update us on the status.

I look forward to receiving your revised manuscript.

Yours sincerely,

Poonam Bheda, PhD
Scientific Editor
Molecular Systems Biology

Reviewer #1:

In this manuscript, Broughton et al. analyze the underlying single-cell dynamics of the antagonistic interaction between antibiotics tetracycline and ciprofloxacin, whereby a combination of the drugs inhibits growth of *E. coli* less than ciprofloxacin alone. Using single-cell microfluidics, the authors find that this population-level effect is caused by a reduced cell death rate by the drug combination. The authors find that cell survival, and therefore antagonism between the drugs, depends on the culture medium conditions. Interestingly, the authors observed that while surviving cells exhibited low levels of the SOS response, dead cells exhibited either low or high levels. Overall, this reviewer found this manuscript interesting and well written. The experiments and the data analysis are solid, and shed light on an interesting phenomenon. The main message of the manuscript is an interesting one, that heterogeneity in cell survival, and not changes in the elongation rate, underlie population-level growth in antagonistic drug interactions. However, I am not fully satisfied with the interpretation of the results, as there are some important questions that are not properly addressed. Similar single-cell experiments where drug exposures resulted in similar phenotypes have been observed before, and it is not clear how these results compare to the existing literature. Ultimately, I am favorable of publication, provided the authors can address the following points:

Major points:

- Similar mother machine experiments have found that drug responses result in growing cells with mid/low response levels and then dead cells with either low or high levels, such as the *E. coli* tetracycline response (PMCID: PMC5857293). This phenotype of dead cells with high expression of the response is a puzzling one. However, the authors do not provide a strong rationale of why over-expressing the SOS response results in cell death, while cells with low SOS expression can go either way. Moreover, from Fig. 4, it seems like the SOS response increases very rapidly, shortly before cell death. It is unclear to me how to connect expression of the SOS response with cell survival, which is a central point of the paper.
- In the discussion, the authors state that despite their current finding that induction of the SOS response was higher in fast-growing cells, they do not expect a higher levels of breaks in those cells. Why should that be the case, since fast-growing cells have higher DNA content? The authors present a higher number of irreparable DSBs in slow-growing cells as an explanation of why high-SOS cells are more abundant in slow growing conditions. However, this explanation seems to contradict the observation that slow-growing cells in gly and glu media are more resistant to CIP treatment.
- Among the low-SOS subpopulation, the authors suggest that cells die through an uncharacterized cellular process. This seems to imply that low SOS is sufficient to prevent death through DNA breaks, and cell death must happen because of other reasons. I am not so convinced by this explanation.
- The death of cells with higher elongation rates is very interesting. The authors say that fast-elongating cells might be unable to down-regulate ribosome, and die as a result. In fact, the elongation rate itself results from the content of functioning ribosomes. Therefore, slow-elongating cells are likely to already have lower protein production. TET up-regulates ribosome, but reduces the pool of functional ones, also avoiding excess protein production. Overall, the relation between media conditions, cell growth, and ribosome production to the suppressive drug interaction is under-explored in this manuscript. The only thing we learn from using different media is that suppression requires fast cell growth.

Minor points:

- Although the authors have covered this in detail in a previous publication, it would be useful to include an introduction to the SOS response to ciprofloxacin in this manuscript as well.
- Plots of cell growth as a function of drug concentration for liquid cultures would also be useful to situate the reader on the population-level growth inhibition of each drug alone in the concentrations used in the experiments.

- Comparing CIP and CIP-TET conditions in Fig. 3D, we can already see that suppression works better under fast growth conditions.
- Comment on the TET impact in gene expression in page 14. If I understood correctly, TET up-regulates ribosome production, decreasing expression of constitutive genes. Since there is more room to up-regulate ribosomes further in slow-growing cells, TET effects would be larger in poorer media. Is that correct? This could be better explained.
- On page 8, the authors state they "measured the single-cell elongation rates, division rates, cell length, and gene expression..." but division rates are not subsequently examined.
- Figure 3 caption does not match the actual figure. It mentions inexistent panels G and H. The main text describes panel 3D as survival fractions for all cells, not only low-SOS ones.
- On page 14, "In the CIP-TET condition, the high-SOS population, surprisingly, did not seem to be affected". Is this compared to CIP alone?
- On page 14, typo: "The low-SOS population did not show not much change".
- Figure 4 caption, the the CIP-TET condition is not mentioned, and the panel labeling seems to be wrong.
- On page 16, "A significant improvement in cell survival was observed for both low and high-SOS cells in all growth conditions...", and then "However, we did not observe a significant improvement in survival for the high-SOS cells, except for the gly medium". Is this statistical significance, in relation to Bliss independence?

Reviewer #2:

This work explores how a combination of antibiotics ciprofloxacin and tetracycline affect individual cells of *E. coli*. The authors show that, while a sub-inhibitory concentration of ciprofloxacin causes significant filamentation and cell death, the addition of tetracycline partially reverses this effect. They show this is more prominent for faster growing cells and for cells with the lower SOS response.

I find this paper very interesting, well written, and timely. I very much like the quantitative approach of the authors and their attention to detail re. the methods, results, and their interpretation. I recommend publication in *Molecular Systems Biology* after the authors have addressed the following issues:

Results:

- Figure 1F,I (and subsequent figures showing the same type of data) shows a very broad distribution of elongation rates, yet the main text claims there was little growth rate heterogeneity in this experiment. I recall that others who looked at elongation rates in the mother machine obtained narrower distributions. Fitting exp. growth to very short intervals could introduce a lot of noise that is just measurement noise, not actual biological variability.
- Figure 3. Something is wrong with the caption of panel D. It seems to overlap with the caption of figure 5
- when reporting changes in the SOS response could you show the ratios e.g. non-drug:CIP, since the numbers representing SOS expression are in arbitrary units anyway.
- Figure 4. There is a mismatch between the letters I,J,K, in the panels and the caption.
- Section 3.5. The first paragraph is confusing. Is this with TET added? You first say that "A significant improvement in cell survival was observed for both low and high-SOS cells" and then (a few sentences below) that "we did not observe a significant improvement in survival for the high-SOS cells".

Microfluidics:

- mean growth rates in the mother machine seem rather low (e.g. glu-aa about 40 min). In bulk cultures I would expect this to be closer to 25-30 min for this medium. What could affect the growth rate in the trenches? Are the trenches rectangular in cross-section or do they also have small side-channels ($\ll 1$ μm) to increase diffusion of nutrients to the cells?
- related to the previous point: was there any bacterial growth in the main channel and tubing delivering the medium? This could affect the growth rate of cells in the mother machine if bacteria upstream of the trenches partially consumed the medium
- it is reported that a peristaltic pump was used for pumping the medium through the main channel but the reported values (1-2 mL/h) are very low for peristaltic pumps. Is this perhaps a mistake, or a special type of pump was used?
- 1% bleach used for sterilization: what chlorine concentration was it (typically this is reported in ppm of chlorine)?
- inlet tubing - what material was used? Did bacteria adhere to it?

Imaging:

- what was the imaging frequency?
- Phototoxicity/photobleaching. Have the authors checked that their protocol does not lead to extensive phototoxicity/bleaching?

The authors use ca. 100 ms camera exposure for both green and red channels, and maximum lamp intensity. Is this also the time the fluorescence shutter remained open? In my experience, it is difficult to go down to below 0.2-0.3s with mechanical shutters, which means the actual sample exposure to light may be much longer than 100 ms.

Data analysis:

- in 2.4.4, did you exponentiate the sum of logarithms to get the survival factor S_f from the CIP and TET survival factors? Please also make "ln" non-italicized since this is not a variable.
- Fig. S1. There is no error bar for the Bliss-independence calculated growth rate. It should be added to see if there is any difference to the CIP-TET case.
- Fig. S4 is not referred to in the text. It is a shame because it is a nice figure that is easy to understand.
- Plate reader measurement. Calculating the growth rate from the exponential fit carries the risk of a systematic error even for the low-OD range used here; the rate will generally come out to be lower than if you count bacteria in a different way (e.g. by plating at ODs from zero up to say 0.1, or using a bioluminescent reporter for very low cell densities). Also, your method of subtracting the background OD of the initial medium may not work reliably because the OD of the blank changes over time, especially if you use TET.

May 7, 2025

Dear Editor,

Thank you for the opportunity to revise our manuscript. We appreciate the reviewers' careful reading of our manuscript and insightful comments. We have thoroughly addressed their comments and are pleased to re-submit our manuscript for your consideration.

Please find attached our Response to Reviewers, with a detailed point-by-point response (in blue) to every reviewer's comment. We have also attached a changes-highlighted copy of the manuscript and appendix, as requested, along with the clean, revised version.

Please could you also add Achille Fraisse's ORCID ID in the author details section (<https://orcid.org/0009-0003-0338-9285>).

We are confident that our revisions have addressed the reviewers' criticisms and greatly strengthened our work. We look forward to your consideration of our revised manuscript.

Yours sincerely,

Prof. Meriem El Karoui

Point by point answer

Reviewer #1:

In this manuscript, Broughton et al. analyze the underlying single-cell dynamics of the antagonistic interaction between antibiotics tetracycline and ciprofloxacin, whereby a combination of the drugs inhibits growth of *E. coli* less than ciprofloxacin alone. Using single-cell microfluidics, the authors find that this population-level effect is caused by a reduced cell death rate by the drug combination. The authors find that cell survival, and therefore antagonism between the drugs, depends on the culture medium conditions. Interestingly, the authors observed that while surviving cells exhibited low levels of the SOS response, dead cells exhibited either low or high levels. Overall, this reviewer found this manuscript interesting and well written. The experiments and the data analysis are solid, and shed light on an interesting phenomenon. The main message of the manuscript is an interesting one, that heterogeneity in cell survival, and not changes in the elongation rate, underlie population-level growth in antagonistic drug interactions. However, I am not fully satisfied with the interpretation of the results, as there are some important questions that are not properly addressed. Similar single-cell experiments where drug exposures resulted in similar phenotypes have been observed before, and it is not clear how these results compare to the existing literature. Ultimately, I am favorable of publication, provided the authors can address the following points:

Major points:

1. Similar mother machine experiments have found that drug responses result in growing cells with mid/low response levels and then dead cells with either low or high levels, such as the *E. coli* tetracycline response (PMCID: PMC5857293). This phenotype of dead cells with high expression of the response is a puzzling one. However, the authors do not provide a strong rationale of why over-expressing the SOS response results in cell death, while cells with low SOS expression can go either way. Moreover, from Fig. 4, it seems like the SOS response increases very rapidly, shortly before cell death. It is unclear to me how to connect expression of the SOS response with cell survival, which is a central point of the paper.

We do not think that overexpression of SOS in the High SOS cells is necessarily the direct cause of cell death. Rather, it is likely a symptom of the underlying cause of death, which we speculate to be an accumulation of irreparable DSBs. The rapid increase in SOS levels (measured by GFP fluorescence per area as a proxy for concentration) is probably due to continued expression from the SOS promoter until cells stop elongating: as the elongation rate slows down, dilution of GFP decreases, leading to higher concentration. We have created a new figure (Figure EV1) to highlight this. It shows that the jump in SOS levels is approximately correlated with

the time of cell death, and for most cells the SOS levels increase rapidly shortly after elongation arrest (some of the delay might be due to GFP maturation time). This is explained further in the results section (second paragraph Section 2.4).

We thank the referee for pointing out article (PMCID: PMC5857293), which is now mentioned in the discussion (third paragraph, Section 3). Indeed, the fact that different phenotypic behaviours lead to different outcomes is directly relevant to our work and reminiscent of the low and high-SOS death we observed. However, we hesitate to make a direct mechanistic comparison because our strain does not express TetA or TetR.

2. In the discussion, the authors state that despite their current finding that induction of the SOS response was higher in fast-growing cells, they do not expect a higher levels of breaks in those cells. Why should that be the case, since fast-growing cells have higher DNA content?

In the discussion, we referred to our previous publication (PMID: 35620827), where we measured the rate of SOS induction using a genetic system that produces DSB only at the replication forks. This is an important distinction from the random nature (in time and chromosomal location) of DNA breaks induced under CIP treatment. With the replication-dependent genetic system, we expected the rate of DNA break formation to depend strongly on the growth rate. However, under CIP treatment, the rate of DNA break formation is likely less dependent on the growth rate, leading to a scenario where DNA repair efficiency between slow and fast-growing cells is more dominant. Although fast-growing cells have a higher DNA content and thus more targets for CIP, we do not necessarily expect frequent high SOS induction because of higher efficiency of DNA repair due to the presence of multiple chromosome copies for homologous recombination, and therefore a lower level of irreparable breaks. This has been elaborated upon in the 2nd paragraph of the discussion (Section 3).

(cont.) The authors present a higher number of irreparable DSBs in slow-growing cells as an explanation of why high-SOS cells are more abundant in slow growing conditions. However, this explanation seems to contradict the observation that slow-growing cells in gly and glu media are be more resistant to CIP treatment.

We realise that the interpretation of the results is not straightforward:

In fast growth (glu-aa), more cells die (64%), but the large majority of them are low SOS (52%). In slow growth (gly), the overall death rate is lower (57%), but many of the cells that die are high SOS (30%), which means that they are more frequent in this condition than in fast growth (12%).

There is, therefore, no contradiction but most likely two different mechanisms at play that have different importance depending on growth conditions.

The data is presented in Figure 4 and we also described this in a sketch in the graphical abstract, which we hope helps simplify the complexity.

3. The authors suggest that cells die in the low-SOS subpopulation through an uncharacterized cellular process. This seems to imply that low SOS is sufficient to prevent death through DNA breaks, and cell death must happen for other reasons. However, I am not convinced by this explanation.

As we have stated in the manuscript, we do not have a clear mechanistic explanation for the death of the low SOS population. Because of their limited SOS induction and higher abundance in glu-aa (when a large part of the chromosome is present in multiple copies because of multi-fork replication), we hypothesise that cellular death is not due to an irreparable break but rather that these cells die through another molecular process.

Prompted by the referee, we re-analysed the data and noticed that the fraction of low-SOS cells that eventually die correlates positively with elongation rate (Figure EV2). This is reminiscent of the higher mass fraction of ribosomes in fast growth predicted by the growth laws (PMID: 21097934). This has led us to suggest that these cells die because of toxicity due to an irreversible impairment of their ribosomes, which would be more frequent when ribosomes are more abundant. For example, the YafO toxin (PMID: 19617347, PMID: 19837801), which is SOS regulated, has been reported to target the ribosomes and could, at least in part, lead to this toxicity. Tetracycline, which also binds the ribosomes, could act as a competitive inhibitor of the toxins thus preventing cell death. This hypothesis is now presented in 4th paragraph of the discussion (section 3)

4. The death of cells with higher elongation rates is very interesting. The authors say that fast-elongating cells might be unable to down-regulate ribosome, and die as a result. In fact, the elongation rate itself results from the content of functioning ribosomes. Therefore, slow-elongating cells are likely to already have lower protein production. TET up-regulates ribosome, but reduces the pool of functional ones, also avoiding excess protein production.

This comment perfectly matches our new hypothesis that high ribosome-containing cells, which are elongating faster, might die more because of a toxin binding to the ribosomes which would be alleviated by competitive binding of TET.

Overall, the relation between media conditions, cell growth, and ribosome production to the suppressive drug interaction is under-explored in this manuscript. The only thing we learn from using different media is that suppression requires fast cell

growth.

We hope that our new analyses and additional discussion help clarify this point.

Minor points:

5. Although the authors have covered this in detail in a previous publication, it would be useful to include an introduction to the SOS response to ciprofloxacin in this manuscript as well.

We have expanded on the SOS response, which is described in the second paragraph of the introduction (Section 1). We included a summary of its regulation, its induction by sub-MIC ciprofloxacin, and its link to toxin-antitoxin systems.

6. Plots of cell growth as a function of drug concentration for liquid cultures would also be useful to situate the reader on the population-level growth inhibition of each drug alone in the concentrations used in the experiments.

Another set of experiments was performed evaluating the bulk doubling rates (OD600 measurements) under each drug individually and in combination at the concentrations tested in the mother machine. This was done in triplicate in the three growth media. This dataset expands on Appendix Fig S1C (initial submission), which is now represented in a separate figure as Appendix Fig S2.

7. Comparing CIP and CIP-TET conditions in Fig. 3D, we can already see that suppression works better under fast growth conditions.

Indeed. We discuss this later in the text, but, since this result is already obvious at this point (Fig 3) we included an explicit statement at the end of the second paragraph of Section 2.3.

8. Comment on the TET impact in gene expression in page 14. If I understood correctly, TET up-regulates ribosome production, decreasing expression of constitutive genes. Since there is more room to up-regulate ribosomes further in slow-growing cells, TET effects would be larger in poorer media. Is that correct? This could be better explained.

Yes, the reviewer's summary is correct. This has been explained in more detail in the manuscript (first paragraph Section 2.4).

9. On page 8, the authors state they "measured the single-cell elongation rates, division rates, cell length, and gene expression..." but division rates are not subsequently examined.

We have now removed mention of single-cell division rates in the manuscript since it is not discussed or presented in the results (second paragraph of Section 2.1).

10. Figure 3 caption does not match the actual figure. It mentions inexistent panels G and H. The main text describes panel 3D as survival fractions for all cells, not only low-SOS ones.

We apologize for the error in labelling, which has now been fixed. References to panels G and H have been removed. Description of panel 3D corrected to match what was described in the main text. The survival fractions plotted in Figure 3D represents all cells.

11. On page 14, "In the CIP-TET condition, the high-SOS population, surprisingly, did not seem to be affected". Is this compared to CIP alone?

Yes, this statement was made in comparison to the CIP-only condition. The sentence has been amended to make this clear (first paragraph of Section 2.4).

12. On page 14, typo: "The low-SOS population did not show not much change".

We thank the reviewer for noting this error. This has been corrected (first paragraph of Section 2.4).

13. Figure 4 caption, the CIP-TET condition is not mentioned, and the panel labeling seems to be wrong.

A reference to the CIP-TET condition has been included in the legend of Figure 4 which is presented in Appendix Figure S9. The panel labelling in the figure legend has also been corrected. Thank you for bringing this to our attention.

14. On page 16, "A significant improvement in cell survival was observed for both low and high-SOS cells in all growth conditions...", and then "However, we did not observe a significant improvement in survival for the high-SOS cells, except for the gly medium". Is this statistical significance, in relation to Bliss independence?

Yes, the latter statement is with reference to Bliss independence. This has been made clear in the text (first paragraph Section 2.5). To be clear, the first statement the reviewer quoted is referencing the CIP-TET conditions compared to CIP treatment alone. This has also been made clearer in the text.

Reviewer #2:

This work explores how a combination of antibiotics ciprofloxacin and tetracycline affect individual cells of *E. coli*. The authors show that, while a sub-inhibitory concentration of ciprofloxacin causes significant filamentation and cell death, the addition of tetracycline partially reverses this effect. They show this is more prominent for faster growing cells and for cells with the lower SOS response.

I find this paper very interesting, well written, and timely. I very much like the quantitative approach of the authors and their attention to detail re. the methods, results, and their interpretation. I recommend publication in *Molecular Systems Biology* after the authors have addressed the following issues:

Results:

15. Figure 1F,I (and subsequent figures showing the same type of data) shows a very broad distribution of elongation rates, yet the main text claims there was little growth rate heterogeneity in this experiment. I recall that others who looked at elongation rates in the mother machine obtained narrower distributions. Fitting exp. growth to very short intervals could introduce a lot of noise that is just measurement noise, not actual biological variability.

The reviewer raises a good point regarding the variability in elongation rates from our mother machine experiments. In the manuscript (paragraph 2.1), we only emphasised the difference between CIP treatment, where some cells stop elongating altogether, and the other conditions where the elongation rates are more homogenous. We have not performed an in-depth analysis of their heterogeneity.

For calculation of the elongation rate, we do not fit to short intervals but to the whole cell cycle to limit noise issues. The data represents 'raw' growth rate trajectories. Though we filter out whole lineages for various reasons (documented in the materials and methods), we do not apply an algorithm to 'clean up' the noise in the individual growth rate trajectories themselves. Others who have presented mother machine elongation rates with narrower distributions have gone further to clean up this noise. However, we note that our distributions are not significantly different than observed in other studies (PMID: 35344432, see supplementary).

16. Figure 3. Something is wrong with the caption of panel D. It seems to overlap with the caption of figure 5

We are unclear what specific issue the referee refers to. The survival fractions in Fig 3D represent *all* cells that have died, while the survival fractions presented in Fig 5 are subdivided by their SOS-level (either low or high).

17. when reporting changes in the SOS response could you show the ratios e.g. non-drug:CIP, since the numbers representing SOS expression are in arbitrary units anyway.

We have created a new figure (Appendix Figure S12) with the requested change. Here we normalised SOS expression by the median of the no-drug control.

18. Figure 4. There is a mismatch between the letters I,J,K, in the panels and the caption.

We apologise for the mistake. This error has now been rectified.

19. Section 3.5. The first paragraph is confusing. Is this with TET added? You first say that "A significant improvement in cell survival was observed for both low and high-SOS cells" and then (a few sentences below) that "we did not observe a significant improvement in survival for the high-SOS cells".

The latter was in reference to the Bliss expectation (comparing CIP-TET to Bliss). The former statement was comparing the entire survival curve of CIP and CIP-TET. We have clarified this in the text (first paragraph of Section 2.5). The second sentence is now: "However, we did not observe a significant suppressive interaction compared to Bliss independence for the high-SOS cells, except for the gly medium".

Microfluidics:

20. mean growth rates in the mother machine seem rather low (e.g. glu-aa about 40 min). In bulk cultures I would expect this to be closer to 25-30 min for this medium. What could affect the growth rate in the trenches? Are the trenches rectangular in cross-section or do they also have small side-channels ($\ll 1$ μm) to increase diffusion of nutrients to the cells?

We performed experiments to measure bulk doubling times. In glu-aa, our measurement is 30 ± 2 min (this medium does not contain nucleotides, which is necessary to reach around 25 minutes of doubling time). This is now presented in Appendix Table S1.

In the mother machine, we measured single-cell division times with a mean of 40.14 min and a median of 35 min for this medium. While bulk and single-cell growth rates are related, they are different growth parameters, so they may not be identical: bulk growth rates are biased towards the fastest growing/dividing cells due to selection (PMID: 26951676, PMID: 29187636) whereas mother machine data does not involve selection (fixed population of cells).

Our mother machine design does not have small side-channels; the trenches are rectangular in cross-section. However, we specifically designed channel lengths to be no longer than 25 μm to limit the impact of diffusion on the growth of mother cells. Moreover, we were careful to ensure that the width and height of the channels did not restrict cell growth.

21. related to the previous point: was there any bacterial growth in the main channel and tubing delivering the medium? This could affect the growth rate of cells in the mother machine if bacteria upstream of the trenches partially consumed the medium

No, we did not observe any bacterial growth in the main channel or tubing upstream of the microfluidic chip. All tubing (and influent containers) was inspected at the end of each experiment for contamination. Experiments where this did occur were discarded. No growth was observed in the main trench of the microfluidic chip. Flow of medium carried all cells into the effluent. As reported in the materials and methods, cell attachment to PDMS was minimised by adding a very low level of tween-20 (0.01% final concentration) to the influent medium.

22. it is reported that a peristaltic pump was used for pumping the medium through the main channel but the reported values (1-2 mL/h) are very low for peristaltic pumps. Is this perhaps a mistake, or a special type of pump was used?]

We used a peristaltic pump with a minimum flow rate of 0.8 mL/hour. The pump used was an Ismatec IPC ISM932D. The reported flow rates in the manuscript are correct.

23. 1% bleach used for sterilization: what chlorine concentration was it (typically this is reported in ppm of chlorine)?

Our 1% bleach solution is equivalent to 10000 ppm chlorine which is within the typical range used for sterilization. This has been added to the Materials and Methods.

24. Inlet tubing - what material was used? Did bacteria adhere to it?

The material of the tubing used was Tygon. We do not observe any bacterial adhesion in the tubing for the duration of the experiments performed. Note that we add a very low level of tween-20 (0.01% final concentration) to the influent medium as mentioned in the Material and Methods. This prevents bacterial adhesion to the tubing and PDMS and is a common medium supplement in mother machine experiments.

Imaging:

25. what was the imaging frequency?

Every 5 minute for glu-aa and glu or every 10 min for gly. This is described in Section 4.4.2 (third paragraph) of the Materials and Methods.

26. Phototoxicity/photobleaching. Have the authors checked that their protocol does not lead to extensive phototoxicity/bleaching? The authors use ca. 100 ms camera

exposure for both green and red channels, and maximum lamp intensity. Is this also the time the fluorescence shutter remained open? In my experience, it is difficult to go down to below 0.2-0.3s with mechanical shutters, which means the actual sample exposure to light may be much longer than 100 ms.

We are using LEDs as the light source and therefore do not use a mechanical shutter thus minimising light exposure. We observe a very small amount of death for the WT with no cipro, which has been reported in all other machine experiments and has been attributed to ageing, although this could be partially due to phototoxicity. This is why we always have a baseline condition of the WT without cipro.

Data analysis:

27. in 2.4.4, did you exponentiate the sum of logarithms to get the survival factor Sf from the CIP and TET survival factors? Please also make "ln" non-italicized since this is not a variable.

Yes, we did exponentiate the sum of logarithms when calculating the survival factors. This has now been made clear in the second paragraph of Section 4.4.4 of the Materials and Methods. All instances of "ln" have been made non-italicized as suggested.

28. Fig. S1. There is no error bar for the Bliss-independence calculated growth rate. It should be added to see if there is any difference to the CIP-TET case.

We thank the reviewer for making this suggestion. Error bars have been added to the Bliss independence calculation in Appendix Fig S2 of the revised manuscript (previously Appendix Fig S1C). Details on how the errors were calculated have been included in the Data Analysis section in the Methods (second paragraph of Section 4.4.4). We have also included a statistical test for significant differences in means between the Bliss and CIP-TET condition which have been added to all figures with a Bliss comparison (Figure 3, Figure 5) and described in the text (Section 4.5 Materials and Methods, Section 2.4 Appendix Supplementary Methods).

29. Fig. S4 is not referred to in the text. It is a shame because it is a nice figure that is easy to understand.

This figure has now been referred to in the methods (Section 4.4.4) and results (Section 2.3 second paragraph) sections. In the revised manuscript it is now called Appendix Figure S5.

30. Plate reader measurement. Calculating the growth rate from the exponential fit carries the risk of a systematic error even for the low-OD range used here; the rate will generally come out to be lower than if you count bacteria in a different way (e.g.

by plating at ODs from zero up to say 0.1, or using a bioluminescent reporter for very low cell densities). Also, your method of subtracting the background OD of the initial medium may not work reliably because the OD of the blank changes over time, especially if you use TET.

We agree with the referee's comment on the poor reliability of estimating growth rates from low OD (<0.1) measurements, particularly from a plate-reader which has its own inherent issues (condensation, evaporation, mixing). We only used the platereader-derived growth rates to confirm the range of concentrations leading to a suppressive interaction to inform the concentration to use in mother machine experiments. For background subtraction we only used wells that did not contain antibiotics.

To compare our mother machine data to population results with high precision we estimated growth rates from automatic OD measurements using OGI-BIO bioreactors which were shown in Appendix Figure S1C (now Appendix Figure S2 in the revised manuscript with more data included) in order to avoid issues inherent to platereaders. Here we ensured fitting to OD's more than 0.05, above which we found that readings were reliably linear.

4th Jul 2025

Manuscript Number: MSB-2024-12561R

Title: Suppression of bacterial cell death underlies the antagonism between ciprofloxacin and tetracycline

Author: James Broughton

Achille Fraise

Meriem El Karoui

Dear Prof El Karoui,

Thank you for the submission of your revised manuscript to Molecular Systems Biology. We have now received the enclosed reports from the referees that were asked to re-assess it. As you will see the reviewers are now globally supportive and I am pleased to inform you that we will be able to accept your manuscript pending the following final amendments:

1) Please format the Data Availability section according to the example below:

"The datasets and computer code produced in this study are available in the following databases:

- Chip-Seq data: Gene Expression Omnibus GSE46748 (<https://www.ncbi.nlm.nih.gov/geo/query/acc.cgi?acc=GSE46748>)

- Modeling computer scripts: GitHub (<https://github.com/SysBioChalmers/GECKO/releases/tag/v1.0>)

- [data type]: [full name of the resource] [accession number/identifier] ([doi or URL or identifiers.org/DATABASE:ACCESSION])"

2) Our journal encourages inclusion of *data citations in the reference list* to directly cite datasets that were re-used and obtained from public databases. Data citations in the article text are distinct from normal bibliographical citations and should directly link to the database records from which the data can be accessed. In the main text, data citations are formatted as follows: "Data ref: Smith et al, 2001" or "Data ref: NCBI Sequence Read Archive PRJNA342805, 2017". In the Reference list, data citations must be labeled with "[DATASET]". A data reference must provide the database name, accession number/identifiers and a resolvable link to the landing page from which the data can be accessed at the end of the reference. Further instructions are available at .

3) We do not allow statements/conclusions with "data not shown". Please show the significant differences between CIP-TET and CIP alone in Figure 3D or its legend or remove this statement from the figure legend.

4) In the Methods, please rename "Methods and Protocols" to "Methods".

5) Please ensure that a statement on whether or not blinding was done is included in the Methods even if no blinding was done. Please also be sure to update the Author Checklist with this information and where it can be found in the manuscript.

6) Please be sure the individual sections of the manuscript in the following order: Title page - Abstract & Keywords - Introduction - Results - Discussion - Methods - Data Availability - Acknowledgements - Disclosure and Competing Interests Statement - References - Figure Legends - Expanded View Figure Legends.

7) For the figures and figure legends, please take care of the following:

- All Figure callouts should be listed sequentially; currently the callouts for Appendix Figure S10 and S13 as well as Appendix Table S6 are missing.

- Please note that the exact p values are not provided in the legends of figures 3A-D, 5A-H

- Please note that information related to n is missing in the legend of figure 3D

8) Tables EV1 and EV2 do not need to be in a zipped folder. Please upload the excel file as Expanded View Content. The title+legend txt file is not needed as it is already included in the separate tab within the excel file.

9) The title page of the Appendix file should contain "Appendix for + manuscript title" and a Table of Contents with the page numbers for each of the listed items (figures and tables should be listed individually); the nomenclature should be Appendix Figure Sx and Appendix Table Sx throughout the manuscript and Appendix PDF (the words "Supporting" and "Supplementary" should not be used in nomenclature).

10) In a routine image check, we noted that images have been reused in the following figures:

- Appendix Figure S6H and Appendix Figure S9A

- Appendix Figure S7H and Appendix Figure S9B

- Appendix Figure S8H and Appendix Figure S9C

Please recheck your appendix file. If the reuse is warranted, then this should be explicitly stated in the corresponding figure legends.

11) For References that have more than 10 authors on a paper, only the first 10 should be listed, followed by "et al." (e.g. for Murray 2022). Please check "Author Guidelines" for more information.

<https://www.embopress.org/page/journal/17574684/authorguide#referencesformat>

12) As part of the EMBO Publications transparent editorial process initiative (see our policy here:

https://www.embopress.org/transparent-process#Review_Process), Molecular Systems Biology will publish online a Peer Review File (PRF) to accompany accepted manuscripts. This file will be published in conjunction with your paper and will include the anonymous referee reports, your point-by-point response and all pertinent correspondence relating to the manuscript. Let us know whether you agree with the publication of the PRF and as here, if you want to remove or not any figures from it prior to publication. Please note that the Authors checklist will be published at the end of the PRF.

13) After your paper is published, we may promote it on social media. If you have any handles or hashtags for Bluesky you would

like included, please let us know.

14) Please provide a point-by-point letter INCLUDING my comments and your detailed responses (as Word file).

I look forward to reading a new revised version of your manuscript as soon as possible.

Yours sincerely,

Poonam Bheda, PhD
Scientific Editor
Molecular Systems Biology

Reviewer #1:

The authors present a revised version of their manuscript analyzing the underlying single-cell dynamics of the antagonistic interaction between antibiotics tetracycline and ciprofloxacin. The additional data and discussions provided in the revised manuscript significantly improve the interpretation of the results. I believe this manuscript is an important contribution to the field, helping our understanding of the heterogeneous dynamics of bacterial cells during antibiotic exposures. I recommend it for publication.

Reviewer #2:

The authors have addressed all my concerns. I now fully support the publication of this work in Molecular Systems Biology.

August 20, 2025

Dear Editor,

Thank you for revising our manuscript and for your patience. We apologise for the delay. You will find the revised manuscript and accompanying documents have been uploaded. Please find attached our response to your comments below (indicated in blue).

We look forward to seeing the published manuscript.

Yours sincerely,

Prof. Meriem El Karoui

Response to Editor's comments:

1) Please format the Data Availability section according to the example below:

"The datasets and computer code produced in this study are available in the following databases:

- Chip-Seq data: Gene Expression Omnibus GSE46748

(<https://www.ncbi.nlm.nih.gov/geo/query/acc.cgi?acc=GSE46748>)

- Modeling computer scripts: GitHub (<https://github.com/SysBioChalmers/GECKO/releases/tag/v1.0>)

- [data type]: [full name of the resource] [accession number/identifier] ([doi or URL or identifiers.org/DATABASE:ACCESSION])"

The data availability section has been formatted as requested.

2) Our journal encourages inclusion of *data citations in the reference list* to directly cite datasets that were re-used and obtained from public databases. Data citations in the article text are distinct from normal bibliographical citations and should directly link to the database records from which the data can be accessed. In the main text, data citations are formatted as follows: "Data ref: Smith et al, 2001" or "Data ref: NCBI Sequence Read Archive PRJNA342805, 2017". In the Reference list, data citations must be labeled with "[DATASET]". A data reference must provide the database name, accession number/identifiers and a resolvable link to the landing page from which the data can be accessed at the end of the reference. Further instructions are available at

<https://www.embopress.org/page/journal/17574684/authorguide#referencesformat>.

We do not use data from other manuscripts and therefore have no data citation.

3) We do not allow statements/conclusions with "data not shown". Please show the significant differences between CIP-TET and CIP alone in Figure 3D or its legend or remove this statement from the figure legend.

We have clarified this in the text of the figure legend for Figure 3D. The p-values and description of the statistical test have been included.

4) In the Methods, please rename "Methods and Protocols" to "Methods".

The Methods section has been renamed as requested.

5) Please ensure that a statement on whether or not blinding was done is included in the Methods even if no blinding was done. Please also be sure to update the Author Checklist with this information and where it can be found in the manuscript.

Blinding was not done. A statement has been added to the Methods and its location indicated in the Authors checklist.

6) Please be sure the individual sections of the manuscript in the following order: Title page - Abstract & Keywords - Introduction - Results - Discussion - Methods - Data Availability - Acknowledgements - Disclosure and Competing Interests Statement - References - Figure Legends - Expanded View Figure Legends.

This has been checked, and no changes are needed.

7) For the figures and figure legends, please take care of the following:

- All Figure callouts should be listed sequentially; currently the callouts for Appendix Figure S10 and S13 as well as Appendix Table S6 are missing.

- Please note that the exact p values are not provided in the legends of figures 3A-D, 5A-H

- Please note that information related to n is missing in the legend of figure 3D

- Appendix figures have been re-ordered to match the order they are called out in the main text. A reference to Appendix Figure S10 (now called Appendix Figure S12) has been included in the third paragraph of the discussion. A reference to Appendix Figure S13 (now called Appendix Figure S2) has been included in the first paragraph of the results. A reference to Appendix Table S6 has been included in the Data Analysis subsection of the Methods.

- The p-values have now been included in the actual figures.

- Number of cells for the survival fractions in Figure 3D has been added to the end of the figure legend. This does not include the Bliss fraction since these were calculated from the means of the antibiotic conditions.

8) Tables EV1 and EV2 do not need to be in a zipped folder. Please upload the excel file as Expanded View Content. The title+legend txt file is not needed as it is already included in the separate tab within the excel file.

Done as requested.

9) The title page of the Appendix file should contain "Appendix for + manuscript title" and a Table of Contents with the page numbers for each of the listed items (figures and tables should be listed individually); the nomenclature should be Appendix Figure Sx and Appendix Table Sx throughout the manuscript and Appendix PDF (the words "Supporting" and "Supplementary" should not be used in nomenclature).

Done. All instances of supplementary removed and replaced with appendix.

10) In a routine image check, we noted that images have been reused in the following figures:

- Appendix Figure S6H and Appendix Figure S9A
- Appendix Figure S7H and Appendix Figure S9B
- Appendix Figure S8H and Appendix Figure S9C

Please recheck your appendix file. If the reuse is warranted, then this should be explicitly stated in the corresponding figure legends.

Figure panels from Appendix Figure S6-8 (now called Appendix Figures S4-6) were re-used in Appendix Figure S9A-C (now called Appendix Figure S11) for better visualisation of SOS expression between lineages that had survived and died. This has now been explicitly stated in the figure legend of Appendix Figure S9 (now called Appendix Figure S11).

11) For References that have more than 10 authors on a paper, only the first 10 should be listed, followed by "et al." (e.g. for Murray 2022). Please check "Author Guidelines" for more information.

<https://www.embopress.org/page/journal/17574684/authorguide#referencesformat>

The reference list formatting has been amended to conform to guidelines. Formatting style in Overleaf (LaTeX) was generated using the msb.bst file which has been uploaded along with the reference list. Each reference has also been carefully checked that all relevant information is included.

12) As part of the EMBO Publications transparent editorial process initiative (see our policy here: https://www.embopress.org/transparent-process#Review_Process), Molecular Systems Biology will publish online a Peer Review File (PRF) to accompany accepted manuscripts. This file will be published in conjunction with your paper and will include the anonymous referee reports, your point-by-point response and all pertinent correspondence relating to the manuscript. Let us know whether you agree with the publication of the PRF and as here, if you want to remove or not any figures from it prior to publication. Please note that the Authors checklist will be published at the end of the PRF.

We agree with the publication of the PRF.

13) After your paper is published, we may promote it on social media. If you have any handles or hashtags for Bluesky you would like included, please let us know.

We do not have one.

Additional comments

- For Tables EV1 and EV2, please move the table legends from a separate tab/sheet to above the table itself.

The requested change has been made.

- Please renumber the Appendix Tables, as Appendix Table S1 seems to be missing

Appendix Table S1 does appear to be there. Please let us know if you still can't find it.

29th Aug 2025

Manuscript number: MSB-2024-12561RR

Title: Suppression of bacterial cell death underlies the antagonism between ciprofloxacin and tetracycline

Dear Prof El Karoui,

Congratulations on an excellent manuscript, I am pleased to inform you that your manuscript has been accepted for publication in Molecular Systems Biology. Thank you for your comprehensive response to referee concerns. It has been a pleasure to work with you to get this to the acceptance stage.

Yours sincerely,

Poonam Bheda, PhD
Scientific Editor
Molecular Systems Biology
